# Infusing Self-Consistency into Density Functional Theory Hamiltonian Prediction via Deep Equilibrium Models

**Zun Wang**
Microsoft Research AI4Science
Beijing, China, 100084
zunwang@microsoft.com

**Chang Liu**
Microsoft Research AI4Science
Beijing, China, 100084

**Nianlong Zou**
State Key Laboratory of Low Dimensional Quantum Physics and Department of Physics,
Tsinghua University, Beijing, 100084, China

**He Zhang**[*]
Xi'an Jiaotong University, Xi'an, China

**Xinran Wei**
Microsoft Research AI4Science
Beijing, China, 100084

**Lin Huang**
Microsoft Research AI4Science
Beijing, China, 100084

**Lijun Wu**
Microsoft Research AI4Science
Beijing, China, 100084

**Bin Shao**
Microsoft Research AI4Science
Beijing, China, 100084
binshao@microsoft.com

## Abstract

In this study, we introduce a unified neural network architecture, the Deep Equilibrium Density Functional Theory Hamiltonian (DEQH) model, which incorporates Deep Equilibrium Models (DEQs) for predicting Density Functional Theory (DFT) Hamiltonians. The DEQH model inherently captures the self-consistency nature of Hamiltonian, a critical aspect often overlooked by traditional machine learning approaches for Hamiltonian prediction. By employing DEQ within our model architecture, we circumvent the need for DFT calculations during the training phase to introduce the Hamiltonian's self-consistency, thus addressing computational bottlenecks associated with large or complex systems. We propose a versatile framework that combines DEQ with off-the-shelf machine learning models for predicting Hamiltonians. When benchmarked on the MD17 and QH9 datasets, DEQHNet, an instantiation of the DEQH framework, has demonstrated a significant improvement in prediction accuracy. Beyond a predictor, the DEQH model is a Hamiltonian solver, in the sense that it uses the fixed-point solving capability of the deep equilibrium model to iteratively solve for the Hamiltonian. Ablation studies of DEQHNet further elucidate the network's effectiveness, offering insights

---

[*]These authors did this work during an internship at Microsoft Research AI4Science.

38th Conference on Neural Information Processing Systems (NeurIPS 2024).

into the potential of DEQ-integrated networks for Hamiltonian learning. We open source our implementation at `https://github.com/Zun-Wang/DEQHNet`.

# 1   Introduction

Density Functional Theory (DFT) is a framework that has revolutionized the study of physical, chemical, and material systems. By providing insights into the electronic structure of matter, DFT has become an indispensable tool for researchers in various scientific fields. Despite its widespread adoption, the computational intensity of DFT poses a significant bottleneck, particularly when applied to large or complex systems.

To navigate the computational challenges of DFT, machine learning (ML) has emerged as a powerful ally, capable of predicting a myriad of molecular and material properties with reduced computational overhead [1, 2, 3, 4, 5, 6, 7, 8, 9, 10, 11, 12, 13, 14, 15, 16, 17, 18, 19, 20, 21]. However, these ML approaches typically require bespoke models for different properties, necessitating extensive retraining or redesign.

The Hamiltonian, a fundamental quantity in DFT, encapsulates the entire energy and dynamics of a system. Properties of molecules and materials can be considered downstream tasks of the Hamiltonian. Therefore, once the Hamiltonian is accurately predicted, a broad spectrum of properties can be subsequently derived. This realization has spurred interest in using ML to predict the Hamiltonian directly [22]. Recent efforts in this direction have seen substantial improvements by incorporating equivariant graph neural networks, which respect the underlying symmetries of physical laws [23, 24, 25, 26].

In the context of DFT, the Hamiltonian is not just another property—it inherently involves self-consistent iterative processes to achieve convergence. Some studies [27, 28, 29, 30] have incorporated this iterative nature into network training by intertwining DFT computations with the loss function during the training of neural networks, and even leveraging unlabeled data to enhance the model's generalizability. Nonetheless, these methods unavoidably rely on DFT iterations to solve the generalized eigenvalue problem, rendering them impractical for larger systems. This is because they integrate DFT iterations within the training process, melding DFT with the training loss. To fundamentally address this issue, a paradigm shift in network architecture is required.

In this work, we introduce DEQH model, combines deep equilibrium models (DEQs) [31] with off-the-shelf neural network models for the prediction of DFT Hamiltonians. DEQs, characterized by their ability to model systems at an equilibrium state, offer a structural innovation that inherently captures the self-consistent nature of the Hamiltonian without the need for iterative DFT calculations during training. By embedding the equilibrium concept into the architecture, our model seeks to directly learn the fixed-point representation of the Hamiltonian, circumventing the computational complexity while maintaining fidelity to the principles of DFT. Our key contributions are:

- The Hamiltonian's iterative qualities are often neglected by standard machine learning approaches for its direct prediction. Our approach integrates DEQs with off-the-shelf ML frameworks, leveraging node features derived from the Hamiltonian and overlap matrix to harness these iterative aspects. Considering computational efficiency and practical applicability, we adopt QHNet as the backbone of the architecture, leading to the development of DEQHNet.

- Traditional machine learning models primarily serve as Hamiltonian predictors, and while recent self-consistency integrating frameworks aim to refine training, they incur high computational costs. DEQH model distinguishes itself by acting fundamentally as a solver, iteratively determining the Hamiltonian with the deep equilibrium model's fixed-point capabilities. This intrinsic self-consistency and computational efficiency render DEQH model a scalable approach for precise eigenstate prediction without a significant increase in complexity.

- We have benchmarked DEQHNet against the MD17 [22] and QH9 [32] datasets, demonstrating that the incorporation of Hamiltonian self-consistency can significantly enhance predictive accuracy.

- We conduct an ablation study on DEQHNet, and present conjectures regarding the efficacy of networks that incorporate DEQs for the task of learning Hamiltonians.

This architectural leap holds the potential to unlock scalable, efficient, and accurate predictions of DFT Hamiltonians, paving the way for rapid advancements in the understanding and design of new materials and molecules.

## 2   Related works

To date, more and more studies have tackled the intricate challenge of employing machine learning (ML) techniques to model the wavefunction directly. The first study in this domain was conducted by Hegde and Bowen [33], who utilized kernel ridge regression to ascertain the Hamiltonian matrix. Building upon this, Schütt *et al.* introduced the SchNOrb neural network architecture [22], which assembles the Hamiltonian matrix of molecules using a block-wise approach based on atom-pair features. Deep Hamiltonian [24] was developed for the prediction of Hamiltonian matrices of periodic systems. PhiSNet [23] leverage the principles from a series of SE(3)-equivariant models to ensure that predictions maintain an exact physical consistency with the orientation of the input structures. DeepH-E3 [25] formulates the DFT Hamiltonian as a function of the structural configuration of the material. This representation inherently upholds Euclidean symmetry, maintaining this invariance seamlessly even when spin-orbit coupling effects are taken into account. QHNet [26] has been meticulously constructed to eliminate 92% of the tensor product operations to enhance the computational efficiency of the model. DeepH-2 [34] is an equivariant local-coordinate transformer which reduces SO(3) convolutions to SO(2) for efficient Hamiltonian prediction.

Additionally, an increasing number of studies have recognized the inherent self-consistent iterative nature of the Hamiltonian's eigenproperties. These works integrate DFT into the training process to enhance network performance by using this self-consistent iterative characteristic. For instance, Ref. [27] introduced a self-consistent loss that utilizes unlabeled data through amortized DFT, assisting the model training and enabling the exploration of a broader chemical space to improve generalization capabilities. Ref. [30] circumvent the creation of datasets by pretraining with DFT iterations and employing an implicit DFT loss, denoted as $E(\cdot)$, as the training loss function. Simultaneously, a similar approach [28] uses a network predicting the Hamiltonian matrix as an input to DFT to obtain energies, which are then used as optimization targets. Ref. [29] has approached the problem by simultaneously predicting both the Hamiltonian matrix and the density matrix, introducing the direct inversion in the iterative subspace (DIIS) error $\epsilon = HDS - SDH$ as a training signal to avoid the costly self-consistent field calculations. Furthermore, this method can estimate the accuracy of the predictions, making it amenable to integrate with active learning strategies.

## 3   Preliminary

**Equivariance**   Consider a mapping function $\mathcal{L} : \mathcal{X} \to \mathcal{Y}$ that transforms inputs from space $\mathcal{X}$ to outputs within space $\mathcal{Y}$. The function $\mathcal{L}$ is deemed $G$-equivariant if it consistently upholds the symmetry induced by a group $G$ in its mappings. Specifically, for every element $g$ belonging to group $G$, the following relation is satisfied:

$$\mathcal{L} \circ D^{\mathcal{X}}(g) = D^{\mathcal{Y}}(g) \circ \mathcal{L}, \tag{1}$$

where $D^{\mathcal{X}}$ and $D^{\mathcal{Y}}$ denote the actions of the group $G$ on the spaces $\mathcal{X}$ and $\mathcal{Y}$, respectively. This condition guarantees that the function $\mathcal{L}$ faithfully translates the symmetry operations performed on the inputs by $G$ into corresponding transformations in the outputs. For additional concepts relevant to this discussion, please refer to the Supplementary Material A.2-A.3.

**DFT Hamiltonian**   The primary goal of most electronic structure methodologies is to accurately solve the electronic Schrödinger equation:

$$\hat{H}_{el}\Psi_{el} = E_{el}\Psi_{el} \tag{2}$$

where $\hat{H}_{el}$ is the electronic Hamiltonian operator, which encapsulates the interactions and dynamics of electrons, $\Psi_{el}$ represents the electronic wavefunction, and $E_{el}$ denotes the ground state energy. Typically, the electronic wavefunction $\Psi_{el}$ is approximated by an antisymmetric combination of molecular orbitals $\{\psi_i(\mathbf{r})\}$ ($\mathbf{r}$ denotes the electronic coordinates). Generally, each molecular orbital $\psi_i$ is formulated as a linear combination of atom-centered basis functions $\{\phi_j(\mathbf{r})\}$:

$$\psi_i(\mathbf{r}) = \sum_j C_{ij}\phi_j(\mathbf{r}). \tag{3}$$

This formulation leads to the matrix equation:

$$\mathbf{HC} = \mathbf{SC}\epsilon \qquad (4)$$

in which the Hamiltonian is represented as a matrix $\mathbf{H}$. The overlap matrix $\mathbf{S}$, with elements $S_{ij} = \int \phi_i(\mathbf{r})\phi_j(\mathbf{r})d\mathbf{r}$, is introduced to account for the non-orthonormality of the basis functions. To obtain the coefficients $C_{ij}$ that define the wavefunction $\Psi_{el}$ and the orbital energies $\varepsilon$, one must solve a generalized eigenvalue problem. More introduction refers to the Supplementary Material A.1.

**Deep equilibrium model**    The deep equilibrium model (DEQ), a class of innovative implicit layer models introduced by Ref. [31], has garnered considerable attention for its remarkable performance across various large-scale vision and natural language processing tasks. These models often rival or even surpass the state of the art established by traditional explicit models, as demonstrated in subsequent studies by Bai *et al.* [35]. The fundamental principle underpinning this methodology is the concept of an implicit layer that converges to a fixed point through an iterative process.

A typical $k$-layer deep network $h : \mathcal{X} \to \mathcal{Y}$ is defined by a stack of layers,

$$\begin{aligned} z_1 &= x, \\ z_{i+1} &= \sigma\left(W_i z_i + b_i\right), \quad i = 1, \cdots, k-1, \\ h(x) &= W_k z_k + b_k. \end{aligned} \qquad (5)$$

To allow for the continuous integration of input information throughout the depth of the model, an input injection mechanism is incorporated across the layers, which involves adding a linear transformation of the input, denoted as $Ux$. Consequently, this augmented model can be characterized as

$$\begin{aligned} z_1 &= x, \\ z_{i+1} &= \sigma\left(W_i z_i + Ux + b_i\right), \quad i = 1, \cdots, k-1, \\ h(x) &= W_k z_k + b_k. \end{aligned} \qquad (6)$$

To model an infinitely deep network ($k \to \infty$), it is observed that the layer values typically converge to a fixed point, $z^* = \sigma\left(W z^* + Ux + b\right)$. The inclusion of input injection $Ux$ in the model is critical due to the equilibrium point's independence from any initial $z_1$ value. Otherwise, the network's output would become paradoxically invariant to the input. Drawing on the insight that deep sequence model layers often converge to a fixed point, the DEQ approach employs root-finding methods to identify these equilibrium points, thereby optimizing the learning process. For a comprehensive explanation of DEQ, please see the Supplementary Material A.7.

## 4    Methods

Our DEQ-based framework, depicted in Fig.1, is designed as a Hamiltonian solver, contrasting with traditional predictors. It inputs the Hamiltonian, overlap matrix, and structural data to yield the Hamiltonian output, which is then recursively processed through DEQ to find its fixed point. Theoretically, the Hamiltonian and structural information alone could suffice, as structural data can function as a conduit within DEQ to maintain fixed point traits, with atomic orbital representations learned from the structure itself (see Sec. 5.3). Nevertheless, we include the overlap matrix, given its role in DFT's iterative equation (Eq. 4), ease of calculation, and similarity in node feature construction.

### 4.1    Diagonal reduction

The matrix element between atom $i$ and $j$ of single-electron operator $\hat{\mathcal{O}} \in \{\hat{H}, \hat{S}\}$ represented in the atomic orbitals $\{\Phi\}$ is

$$\mathbf{T}_{i\mu,j\nu} = \langle \Phi_i^\mu | \hat{\mathcal{O}} | \Phi_j^\nu \rangle, \qquad (7)$$

where $\mu := (n_1, l_1, m_1)$ and $\nu := (n_2, l_2, m_2)$ are the basis functions in the set of atomic orbitals centered at each atom, respectively. The $n$, $l$, and $m$ are principal, azimuthal, and magnetic quantum numbers, respectively. Refer to OrbNet-Equi [36], for each diagonal block of $\mathbf{T}$, i.e., $\mathbf{T}_{AA}$, defined for an on-site atom pair $(A, A)$, there exists a set of $T$-independent coefficients $Q_{nlm}^{\mu\nu}$ such that the following linear transformation $\psi$,

$$\mathbf{h}_A^l := \sum_{\mu,\nu} T_{AA}^{\mu,\nu} Q_{nlm}^{\mu,\nu} \qquad (8)$$

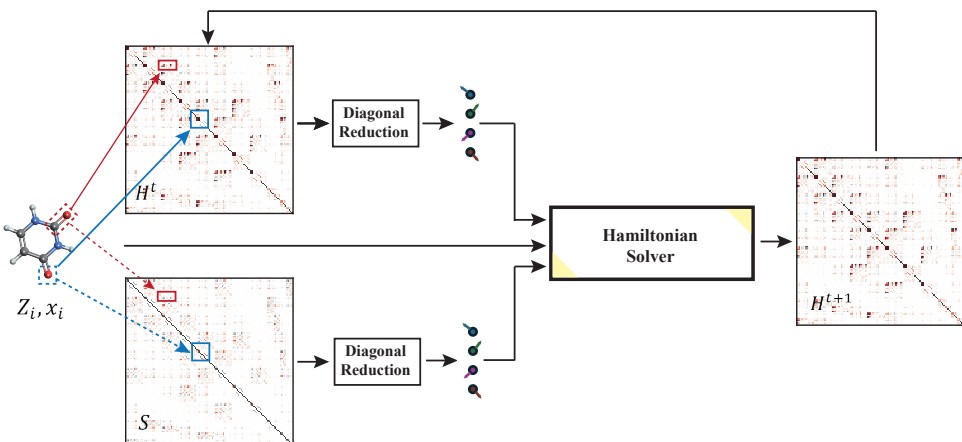

Figure 1: Schematic of the DEQH model. A Hamiltonian solver must be engineered to concurrently process structural information, the Hamiltonian, and the overlap matrix, and to output the subsequent Hamiltonian iteration for DEQ convergence. Within the Hamiltonian solver, the module handling structural information is termed injection, while the remaining components are collectively referred to as filter.

is injective and satisfy equivariance. The existence of $\mathbf{Q}$ could be proved by Wigner–Eckart theorem. For the sake of computational feasibility, a physically-motivated scheme is employed to tabulate $\mathbf{Q}$ and produce order-1 equivariant embeddings $\mathbf{h}_A$, using *on-site 3-index overlap integrals* $\tilde{\mathbf{Q}}$:

$$
\begin{aligned}
\tilde{Q}_{nlm}^{\mu,\nu} &:= \tilde{Q}_{nlm}^{n_1,l_1,m_1;n_2,l_2,m_2} \\
&= \int_{\mathbf{r}\in\mathbb{R}^3} (\Phi_A^{n_1,l_1,m_1}(\mathbf{r}))^* \Phi_A^{n_2,l_2,m_2}(\mathbf{r}) \tilde{\Phi}_A^{n,l,m}(\mathbf{r}) d\mathbf{r}
\end{aligned}
\tag{9}
$$

where $\Phi_A$ are the atomic orbital basis, and $\tilde{\Phi}_A$ are auxiliary Gaussian-type basis functions. $\tilde{\mathbf{Q}}$ adheres to equivariance constraints due to its relation to SO(3) Clebsch-Gordan coefficients $C_{l_1m_1;l_2m_2}^{lm} \propto \int_{\mathbf{r}\in\mathbb{S}^2} Y_{l_1m_1}(\mathbf{r})Y_{l_2m_2}(\mathbf{r})Y_{lm}^*(\mathbf{r})d\mathbf{r}$. Consequently, the role of $\mathbf{Q}$ is to couple the indices $l_1$ and $l_2$ of the Hamiltonian matrix elements, resulting in features of order $l$. Details could be found in Supplementary Material A.4-A.6.

## 4.2 Hamiltonian solver

In this paper, we have chosen to use QHNet [26] because of its versatility and adaptability. QHNet is capable of supporting multiple molecules simultaneously. Its matrix prediction module is designed to predict matrices of varying sizes for different molecules [32], making it a more flexible and robust choice for our study. As illustrated in Fig. 2, we present the comprehensive architecture of DEQHNet. To effectively incorporate the self-consistent nature of the Hamiltonian through the DEQ mechanism within the network architecture, it is imperative that the network's filters accept the Hamiltonian as input. Additionally, structural information must be used as an injection to establish dependencies at the fixed point. Consequently, this necessitates the design of a Hamiltonian solver tailored to these requirements. This solver will enable the network to iteratively refine its predictions, ensuring that the output Hamiltonian is self-consistent and in line with the structural information provided, thereby harnessing the full potential of the DEQ framework in electronic structure modeling.

**Injection** As indicated in the boxed area of Fig. 2(a), the section highlighted represents the injection component of DEQHNet. Within this component, the atomic number $\{Z\}$ is input as the node feature, while the relative position vectors $\vec{r}_{ij}$ are processed through Radial Basis Functions (RBF) and spherical harmonics to construct the edge features that are invariant and equivariant, respectively. This design ensures that the crucial chemical information, such as the type of elements and their spatial relations, are effectively encoded into the network, allowing DEQHNet to accurately capture the nuances of molecular structures and interactions.

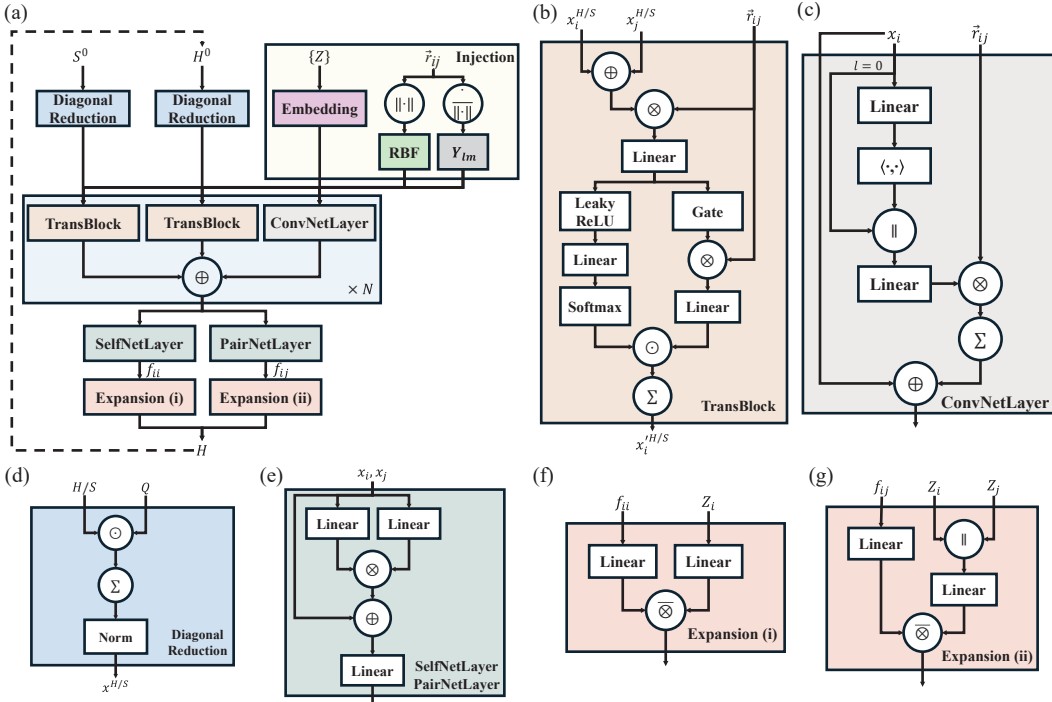

Figure 2: A schematic diagram of DEQHNet. (a) The overall architecture of DEQHNet, with the injection mechanism encapsulated within the yellow frame and the remainder designated as the filter. (b) TransBlock. Integrates the Hamiltonian and overlap matrix with structural information (relative position vectors). (c) ConvNetLayer. Responsible for processing structural information. (d) Diagonal Reduction. Transforms the Hamiltonian and overlap matrix into equivariant node features. (e)-(g) Couple node features to form diagonal and off-diagonal blocks of the Hamiltonian.

**Filter**    Our design draws inspiration from QHNet [26] and Equiformer [15] to handle structural information, as well as node features constructed from the Hamiltonian and the overlap matrix. As depicted in Fig. 2 (d), the Hamiltonian and overlap matrix are initially processed using the previously described diagonal reduction method to construct node features. It is noteworthy that the initial value of the Hamiltonian provided to the Hamiltonian solver is set as the identity matrix $\mathbf{I}$, a deliberate choice to preserve the Hermiticity of the Hamiltonian. These features are then normalized as $x_i^{H/S(l_T)}$ before being fed into the module illustrated in Fig. 2 (b). Within this module, the equivariant node features are combined with the equivariant relative position vectors through a tensor product, resulting in the generation of new node features,

$$
\begin{aligned}
h_{ij}^{H/S,(l_o)} &= \left( \text{Linear}(x_i^{H/S,(l_T)}) + \text{Linear}(x_j^{H/S,(l_T)}) \right) \otimes Y_{l_i m}(\vec{r}_{ij}), \\
a_{ij} &= \text{Softmax}\left( W\text{LeakyReLU}(h_{ij}^{H/S,l_o=0}) \right), \\
v_{ij} &= \text{Linear}\left( h_{ij}^{H/S,(l_o)} \otimes Y_{l_i m}(\vec{r}_{ij}) \right), \\
\tilde{x}_i^{H/S,l} &= \sum_{j \in \mathcal{N}_i} a_{ij} v_{ij}.
\end{aligned}
\tag{10}
$$

As depicted in Fig. 2 (c), the filtering module within the Tensor Field Network (TFN) layer regulates the influence of messages coming from other nodes. In processing structural features, the inner product $I_{ij}^l$ of the $l$-order irreducible representation on a pair is calculated as

$$
I_{ij}^l = \langle \text{Linear}(x_i^l), \text{Linear}(x_j^l) \rangle.
\tag{11}
$$

Subsequently, the pairwise cosine similarity of the irreducible representations, along with the 0-order irreducible expression, are fed into a MLP to compute attention scores $a_{ij}$. The calculation of the message $m_{ij}$ from node $j$ to node $i$ is constructed through a tensor product of the transformed relative

position vector $\hat{r}_{ij}$ and the feature vector $\hat{x}_{jc}^{l_{\text{in}}}$,

$$m_{ij}^{l_{\text{out}}} = \sum_{l_f, l_{\text{in}}} a_{ijc}^{(l_{\text{in}}, l_f, l_{\text{out}})} R_c^{(l_{\text{in}}, l_f, l_{\text{out}})}(r_{ij}) Y_{l_f m}(\hat{r}_{ij}) \otimes \hat{x}_{jc}^{l_{\text{in}}}. \tag{12}$$

The output irreducible representation $\tilde{x}_i^l$ is then obtained by aggregating the messages $m_{ij}^l$ and self-connections to form the updated feature:

$$\tilde{x}_i^l = \text{Linear}\left( \hat{x}_i^l + \sum_{j \in \mathcal{N}_i} m_{ij}^l \right). \tag{13}$$

Subsequently, the node features $\tilde{x}_i^{H/S,l}$ constructed from the Hamiltonian and overlap matrix are merged with the node features $\tilde{x}_i^l$ containing structural information. This amalgamation is employed to construct the iteratively refined Hamiltonian for the solver. As shown in Fig. 2(e), the block is designed to harness the tensor product between nodes to separately construct on-site features and pairwise features. These are then channeled into the module depicted in Fig. 2(f)-(g), where the inverse operation of the tensor product [26], i.e. the tensor expansion operation,

$$\bar{\otimes}_{l_1, m_1; l_2, m_2}^{l_3} w^{l_3} = \sum_{m_3 = -l_3}^{l_3} C_{l_1, m_1; l_2, m_2}^{l_3, m_3} w_{m_3}^{l_3}, \tag{14}$$

is applied to construct the Hamiltonian for the next iteration step of the solver.

We have developed a complete Hamiltonian solver that seamlessly integrates with the Differentiable Equation (DEQ) method for fixed-point iterations, enforcing the Hamiltonian's inherent self-consistency. This solver iteratively refines the Hamiltonian, mirroring the self-consistency principle in physics. Utilizing the DEQ framework, it dynamically adjusts parameters to achieve convergence, yielding a Hamiltonian that precisely captures the interactions. This approach not only improves accuracy but also aligns with the iterative nature of electronic structure calculations.

### 4.3 Mathematical formulations of Hamiltonian solver and predictor

As illustrated in Fig. 3, the DEQH model operates as a Hamiltonian solver, learning the iterative process $H^* = f(H^*, Z, R)$. In contrast, models like QHNet function as predictors, using the formula $H = f(Z, R)$, where $Z$ and $R$ represent atomic numbers and coordinates, respectively. The function $f$ typically involves a neural network, and $H$ denotes the Hamiltonian. The superscript $H^*$ indicates the converged Hamiltonian.

In practice, if the dataset includes overlap matrices, these can also be incorporated into the network input, transforming them into equivariant node features akin to the Hamiltonian. Even if overlap matrices are not

(a)

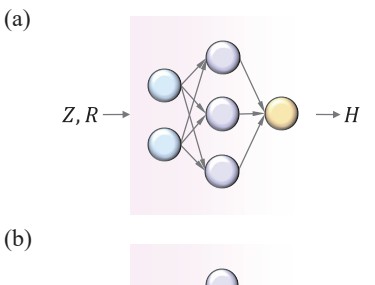

(b)

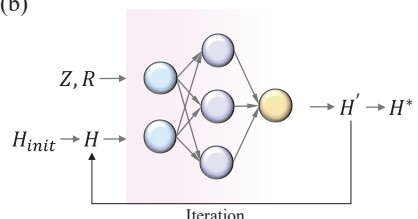

Figure 3: (a) The network predicts the Hamiltonian using atomic number $Z$ and coordinates $R$ as inputs, outputting the molecular Hamiltonian $H$. (b) The DEQH model also includes an input for $H$ and iteratively refines the Hamiltonian until reaching a fixed-point solution $H^*$, hence we refer to this block as the Hamiltonian solver.

initially present, they can be easily calculated during data preprocessing using $Z$ and $R$. The overlap matrix provides a wealth of detailed information, enhancing the model's input. Consequently, the equations for the DEQH and the modified Hamiltonian predictor become $H^* = f(H^*, Z, R, S)$ and $H = f(Z, R, S)$, respectively, where $S$ is the overlap matrix.

## 5 Results

We evaluated the performance of DEQHNet by applying it to the MD17 [22] and QH9 [32] datasets. We have further explored the convergence behavior of DEQHNet. Additionally, we conducted

an ablation study to isolate and understand the specific impacts of the DEQ mechanism and the architectural design on our method's effectiveness. For descriptions of the metrics and results pertaining to acceleration ratio, please consult the Supplementary Material A.8-A.9.

## 5.1 MD17

MD17 dataset [22] encompasses an array of molecular properties, including structures, energies, forces, and Hamiltonians—specifically Fock or Kohn-Sham matrices—as well as overlap matrices pertaining to molecules such as water, ethanol, malondialdehyde, and uracil. All computational analyses were executed at the PBE/def2-SVP level of theory [37, 38], employing the ORCA electronic structure software [39]. Table 1 shows DEQHNet's performance on the MD17 dataset against QHNet. Except for water, DEQHNet achieves lower mean absolute error (MAE) in Hamiltonian predictions for three molecules, a benefit likely from more extensive training data that prevents overfitting. The predicted Hamiltonian improves orbital coefficients and the MAE for orbital energies has risen, which indicates that lower Hamiltonian MAE doesn't necessarily correlate with better orbital energy predictions.

Table 1: Comparison of the MAEs between QHNet and DEQHNet trained on MD17 dataset. The optimal and the second best values are highlighted in **bold** and underlining, respectively. It should be noted that due to the design of DeepH [24], the data used is re-labeled by OpenMX [40], which may lead to different results compared to other models. Furthermore, considering the high training cost of PhiSNet (ori) [23], PhiSNet (reproduce) results from the QHNet [26] article are provided for a more balanced and comprehensive comparison.

| Dataset | Model | $H$ [$10^{-6}E_h$] ↓ | $\epsilon$ [$10^{-6}E_h$] ↓ | $\psi$ [$10^{-2}$] ↑ |
|---|---|---|---|---|
| Water | SchNOrb | 165.4 | 279.3 | **100.00** |
| | PhiSNet (ori) | 17.59 | 85.53 | **100.00** |
| | PhiSNet (reproduce) | 15.67 | - | 99.94 |
| | DeepH | 38.51 | - | - |
| | QHNet | **10.79** | **33.76** | 99.99 |
| | DEQHNet | 36.07 | 335.86 | 99.99 |
| Ethanol | SchNOrb | 187.4 | 334.4 | **100.00** |
| | PhiSNet (ori) | **12.15** | **62.75** | **100.00** |
| | PhiSNet (reproduce) | 20.09 | 102.04 | 99.81 |
| | DeepH | 22.09 | - | - |
| | QHNet | 20.91 | 81.03 | 99.99 |
| | DEQHNet | 18.73 | 106.94 | **100.00** |
| Malonaldehyde | SchNOrb | 191.1 | 400.6 | 99.00 |
| | PhiSNet (ori) | **12.32** | **73.50** | **100.00** |
| | PhiSNet (reproduce) | 21.31 | 100.60 | 99.89 |
| | DeepH | 20.10 | - | - |
| | QHNet | 21.52 | 82.12 | 99.92 |
| | DEQHNet | 17.97 | 93.79 | 99.90 |
| Uracil | SchNOrb | 227.8 | 1760 | 90.00 |
| | PhiSNet (ori) | **10.73** | **84.03** | **100.00** |
| | PhiSNet (reproduce) | 18.65 | 143.36 | 99.86 |
| | DeepH | 17.27 | - | - |
| | QHNet | 20.12 | 113.44 | 99.89 |
| | DEQHNet | 15.07 | 107.49 | 99.89 |

## 5.2 QH9

The QH9 dataset [32], crafted using PySCF [41], is split into QH9-stable and QH9-dynamic. QH9-stable contains 130,831 Hamiltonian matrices from QM9 [42, 43], while QH9-dynamic has 100-geometry trajectories for 999 molecules. DFT calculations employ a grid level of 3, SCF convergence with a $10^{-13}$ tolerance, and a $3.16 \times 10^{-5}$ gradient limit. The B3LYP functional and def2SVP GTO

basis set are used, with DIIS aiding SCF convergence. QH9-dynamic's molecular dynamics were run at 300K in an NVE ensemble. The QH9-dynamic-100k subset features 0.12 fs step trajectories, taken every 10th step for 1,000 steps. Table 2 displays the results of DEQHNet on the QH9 dataset, where a significant reduction in the MAE of the Hamiltonian across both the QH9-stable and QH9-dynamic subsets can be seen. The decrease in MAE is particularly striking for the QH9-dynamic subset, where it nearly halves. The fidelity of the orbital coefficients, derived from the Hamiltonians predicted by DEQHNet, reached over 99% accuracy. This is attributed to the fact that the overlap matrix and the Hamiltonian offer significant insights into the basis set, thereby facilitating the expression of orbital coefficients. Similar to the trend observed with the MD17 dataset, there is no direct positive correlation between the MAE of the Hamiltonian and the orbital energies. Additional experiments was performed to elucidate this situation, with details provided in the Supplementary Material A.10.

Table 2: Comparison of the MAEs between QHNet and DEQHNet trained on QH9 dataset. The optimal values are highlighted in **bold**.

| Dataset | Model | $H$ [$10^{-6}E_h$] $\downarrow$ | | | $\epsilon$ [$10^{-6}E_h$] $\downarrow$ | $\psi$ [$10^{-2}$] $\uparrow$ |
| | | diagonal | non-diagonal | all | | |
| --- | --- | --- | --- | --- | --- | --- |
| QH9-stable-id | QHNet | 111.21 | 73.68 | 76.31 | **798.51** | 95.85 |
| | DEQHNet | **96.43** | **58.75** | **61.42** | 4383.10 | **99.84** |
| QH9-stable-ood | QHNet | 111.72 | 69.88 | 72.11 | **644.17** | 93.68 |
| | DEQHNet | **81.01** | **51.66** | **53.23** | 5657.07 | **99.80** |
| QH9-dynamic-geo | QHNet | 149.62 | 92.88 | 96.85 | **834.47** | 94.45 |
| | DEQHNet | **84.97** | **60.04** | **62.14** | 1864.06 | **99.92** |
| QH9-dynamic-mol | QHNet | 416.99 | 153.68 | 173.92 | 9719.58 | 79.15 |
| | DEQHNet | **210.76** | **97.18** | **105.80** | **4625.88** | **99.80** |

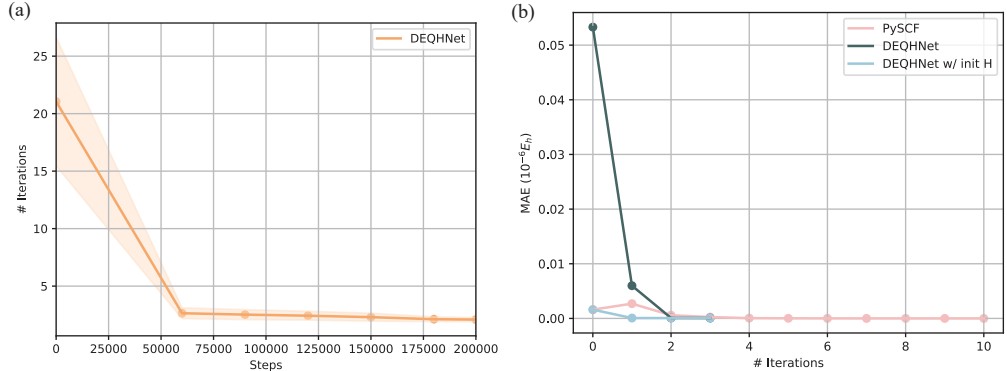

Figure 4: (a) The variation in the number of DEQ iterations within DEQHNet as a function of training steps. For this analysis, 50 random configurations were selected from the uracil test set, and the iteration counts were tallied after performing inference with DEQHNet checkpoints saved during the training process. (b) The change in Hamiltonian MAE with respect to iteration count for a randomly chosen molecule from the QH9-dynamics-geo test set, comparing the self-consistent field (SCF) iterations using PySCF, DEQHNet inference, and DEQHNet inference initialized with Hamiltonians guessed by PySCF.

## 5.3 On the convergence of DEQHNet

We further explored DEQHNet through additional experiments. As depicted in Fig. 4(a), we inferred 50 uracil test set configurations using DEQHNet checkpoints from various training phases. Initially, the model required many iterations to achieve convergence but soon stabilized at two or three iterations, showing quick adaptation to the Hamiltonian's iterative solution. In comparison, PySCF calculations for these configurations, which are structurally similar due to their molecular dynamics origin, consistently converged at 15 SCF iterations. Initially, DEQHNet surpassed this number but soon settled at fewer iterations. This suggests that DEQHNet learns a distilled version of the DFT

solver, and unlike PySCF, which considers energy for convergence, it uses Hamiltonian differences for this purpose.

Furthermore, we randomly selected a molecule from the QH9-dynamic-geo test set for additional analysis. In Fig. 4(b), we plotted the MAE of the Hamiltonian at each iteration during the PySCF computation and the DEQHNet inference relative to the true Hamiltonian. DEQHNet starts with an identity matrix as the initial Hamiltonian, leading to a high initial error, while PySCF employs MINAO density [41]-the superposition of orbitals of each atom in isolation-for its initial guess, resulting in a lower starting MAE. DEQHNet typically stabilizes within three iterations, unlike PySCF, which can have initial optimization fluctuations. By initiating DEQHNet with PySCF's starting Hamiltonian [44], we noted improved stability in convergence. This suggests that DEQHNet's iterative refinement can benefit from a more accurate initial guess, enhancing the stability and reliability of convergence, an important aspect in electronic structure computations.

## 5.4 Ablation studies

To evaluate DEQHNet's efficacy, we conducted ablation studies with two model variants: one excluding the overlap matrix from DEQHNet and another enhancing the original QH-Net with overlap matrix features. As depicted in Fig. 5, DEQHNet's performance dips without the overlap matrix, while the upgraded QHNet shows a reduction in MAE for Hamiltonian prediction but doesn't surpass DEQHNet. The results suggest that both models can infer atomic orbital details from molecular structures and the Hamiltonian labels. The overlap matrix, however, seems to expedite the network's acquisition of atomic orbital information, aiding Hamiltonian construction. Supplying the overlap matrix equips the network with explicit atomic orbital data, facilitating the learning of the effective potential from structural inputs.

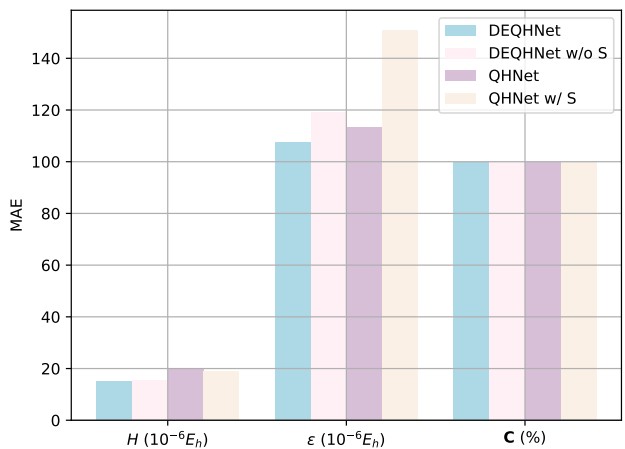

Figure 5: Ablation study for MAE of Hamiltonian $H$, orbital energy $\varepsilon$, and orbital coefficients $\mathbf{C}$ for DEQHNet, DEQHNet without the overlap matrix as an input, QHNet, and QHNet with the addition of the overlap matrix as an input, respectively.

Without this data, the network must align with the Hamiltonian's basis while learning atomic orbital information through loss optimization.

## 6  Conclusion

In essence, the DEQH model fuses DEQs with Hamiltonian learning to obviate the need for iterative DFT computations during the training stage, which are typically required to introduce the self-consistent properties of the Hamiltonian. Our model architecture is meticulously designed to directly capture the fixed-point representation of the Hamiltonian, capitalizing on its intrinsic iterative characteristics to accurately reflect the fundamental principles of molecular systems. This innovative strategy propels the DEQH model beyond the traditional confines of predictive machine learning models, transforming it into an iterative solver that refines Hamiltonians. This paradigm shift ushers in a new epoch of computational efficiency and scalability, enabling the model to adeptly process voluminous datasets and master the complexities of elaborate systems. The DEQH model methodology is a universal one, readily adaptable to established machine learning models for predicting Hamiltonians. With QHNet as the backbone, our DEQHNet has exhibited enhanced predictive accuracy in empirical evaluations on the MD17 and QH9 datasets. We have further delved into the convergence behavior of DEQHNet to elucidate the effectiveness. Our ablation study further reinforces the viability of merging DEQs with neural networks for efficient Hamiltonian learning.

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

# A Supplementary Material

## A.1 Electronic structures

The Hamiltonian for a system composed of electrons and nuclei is given by:

$$\hat{H} = -\frac{\hbar^2}{2m_e}\sum_i \nabla_i^2 + \sum_{i,I}\frac{Z_I e^2}{|\mathbf{r}_i - \mathbf{R}_I|} + \frac{1}{2}\sum_{i \neq j}\frac{e^2}{|\mathbf{r}_i - \mathbf{r}_j|} - \sum_I \frac{\hbar^2}{2M_I}\nabla_I^2 + \frac{1}{2}\sum_{I \neq J}\frac{Z_I Z_J e^2}{|\mathbf{R}_I - \mathbf{R}_J|}, \quad (15)$$

where electrons are represented by lowercase subscripts, while nuclei, characterized by their charge $Z_I$ and mass $M_I$, are denoted by uppercase subscripts.

In the general Hamiltonian, the inverse mass of the nuclei, denoted as $1/M_I$, stands out as the only term that can be considered "small." This allows to establish a perturbation series based on this parameter, which holds broad applicability across the fully interacting electron-nuclei system. Initially, by setting the nuclear mass to infinity, we can disregard the kinetic energy of the nuclei. This leads us to the Born-Oppenheimer or adiabatic approximation. This approximation proves highly effective for various applications, such as calculating nuclear vibration modes in most solids.

With the omission of nuclear kinetic energy, the fundamental Hamiltonian for the theory of electronic structure simplifies to:

$$\hat{H} = \hat{T} + \hat{V}_{\text{ext}} + \hat{V}_{\text{int}} + E_{II}. \quad (16)$$

Adopting Hartree atomic units where $\hbar = m_e = e = 4\pi\epsilon_0 = 1$, we can express the kinetic energy operator for the electrons, $\hat{T}$ as:

$$\hat{T} = \sum_i -\frac{1}{2}\nabla_i^2. \quad (17)$$

The external potential acting on the electrons due to the nuclei, $\hat{V}$ext, is given by:

$$\hat{V}_{\text{ext}} = \sum_{i,I} V_I(|\mathbf{r}_i - \mathbf{R}_I|). \quad (18)$$

$\hat{V}_{\text{int}}$ encompasses the electron-electron interactions,

$$\hat{V}_{\text{int}} = \frac{1}{2}\sum_{i \neq j}\frac{1}{|\mathbf{r}_i - \mathbf{r}_j|}, \quad (19)$$

which quantifies the electron-electron Coulomb repulsion, a fundamental aspect of electronic interactions. The last term, $E_{II}$, represents the classical interactions between nuclei themselves and encompasses any additional contributions to the system's total energy that are not directly related to the description of electron behavior. In this model, the influence of nuclei on the electrons is integrated into a fixed "external" potential. This formulation remains applicable even when the straightforward nuclear Coulomb interaction is substituted with a pseudopotential, which accounts for core electron effects, albeit these potentials are "non-local".

There are two basic independent-particle approaches that may be classified as "non-interacting" and "Hartree-Fock" [45]. They are similar in that each assumes the electrons are uncorrelated except that they must obey the exclusion principle. However, they are different in that Hartree-Fock includes the electron-electron Coulomb interaction in the energy, while neglecting the correlation that is introduced in the true wavefunction due to those interactions. In general, "non-interacting" theories have some effective potential that incorporates some effect of the real interaction, but there is no interaction term explicitly included in the effective Hamiltonian. This approach is often referred to as "Hartree" or "Hartree-like," after D. R. Hartree who included an average Coulomb interaction in a rather heuristic way. More to the point of modern calculations, all calculations following the Kohn-Sham method involve a non-interacting Hamiltonian with an effective potential chosen to incorporate exchange and correlation effects approximately.

According to our comprehensive definition, calculations for non-interacting electrons necessitate solving a Schrödinger-like equation, represented as:

$$\hat{H}_{\text{eff}}\psi_i(\mathbf{r}) = \left[-\frac{\hbar^2}{2m_e}\nabla^2 + V_{\text{eff}}^{\sigma}(\mathbf{r})\right]\psi_i^{\sigma}(\mathbf{r}) = \epsilon_i^{\sigma}\psi_i^{\sigma}(\mathbf{r}), \quad (20)$$

where $V_{\text{eff}}^{\sigma}(\mathbf{r})$ denotes an effective potential influencing each electron of spin $\sigma$ at position $\mathbf{r}$. To determine the ground state for a collection of non-interacting electrons, one must populate the lowest eigenstates, adhering to the exclusion principle. If the Hamiltonian lacks spin-dependence, the spin states, both up and down, are considered degenerate, effectively doubling the count for these states. The occupation of higher energy eigenstates characterizes the excited states. The primary rationale for employing such independent-particle equations for electrons in materials is founded on density functional theory.

At finite temperatures, it is straightforward to utilize the general formulas of statistical mechanics to demonstrate that the equilibrium distribution of electrons conforms to the Fermi-Dirac (or Bose-Einstein) distribution for occupation numbers as a function of energy. The expectation value, aggregates over many-body states $\Psi$, each defined by a set of occupation numbers $n_i^{\sigma}$ for individual particle states with energies $\epsilon_i^{\sigma}$. With each $n_i^{\sigma}$ being either 0 or 1 and the sum $\sum_i n_i^{\sigma} = N^{\sigma}$, it can be shown that:

$$\langle \hat{O} \rangle = \sum_{i,\sigma} f_i^{\sigma} \langle \psi_i^{\sigma} | \hat{O} | \psi_i^{\sigma} \rangle, \tag{21}$$

where $\langle \psi_i^{\sigma} | \hat{O} | \psi_i^{\sigma} \rangle$ represents the expectation value of the operator $\hat{O}$ for the one-particle state $\psi_i^{\sigma}$, and $f_i^{\sigma}$ indicates the probability of finding an electron in state $i$, spin $\sigma$. Specifically, the Fermi-Dirac distribution is given by:

$$f_i^{\sigma} = \frac{1}{e^{\beta(\epsilon_i^{\sigma} - \mu)} + 1}, \tag{22}$$

where $\mu$ represents the Fermi energy or the chemical potential of the electrons. The system's energy, for instance, is the weighted sum of the energies of non-interacting particles:

$$E(T) = \langle \hat{H} \rangle = \sum_{i,\sigma} f_i^{\sigma} \epsilon_i^{\sigma}. \tag{23}$$

Similar to the general many-body scenario, a single-body density matrix operator can be defined:

$$\hat{\rho} = \sum_{i,\sigma} |\psi_i^{\sigma}\rangle f_i^{\sigma} \langle \psi_i^{\sigma}|, \tag{24}$$

where the expectation value $\langle \hat{O} \rangle$ is equal to $\text{Tr}(\hat{\rho}\hat{O})$. For a specific representation involving spin and position, $\hat{\rho}$ is expressed as:

$$\rho(\mathbf{r}, \sigma; \mathbf{r}', \sigma') = \delta_{\sigma,\sigma'} \sum_i \psi_i^{\sigma(\mathbf{r})*} f_i \psi_i^{\sigma}(\mathbf{r}'), \tag{25}$$

where the density is the diagonal component:

$$n^{\sigma}(\mathbf{r}) = \rho(\mathbf{r}, \sigma; \mathbf{r}, \sigma) = \sum_i f_i^{\sigma} |\psi_i^{\sigma}(\mathbf{r})|^2. \tag{26}$$

The Hartree–Fock method is another fundamental approach in many-particle theory, which involves constructing a properly antisymmetrized determinant wavefunction for a predefined number, $N$, of electrons. The goal is to find a single determinant that minimizes the total energy of the fully interacting Hamiltonian. In cases where there is no spin-orbit interaction, this wavefunction, denoted as $\Phi$, can be expressed as a Slater determinant:

$$\Phi = \frac{1}{\sqrt{N!}} \begin{vmatrix} \phi_1(r_1, \sigma_1) & \phi_1(r_2, \sigma_2) & \phi_1(r_3, \sigma_3) & \cdots \\ \phi_2(r_1, \sigma_1) & \phi_2(r_2, \sigma_2) & \phi_2(r_3, \sigma_3) & \cdots \\ \phi_3(r_1, \sigma_1) & \phi_3(r_2, \sigma_2) & \phi_3(r_3, \sigma_3) & \cdots \\ \vdots & \vdots & \vdots & \ddots \end{vmatrix}, \tag{27}$$

where each $\phi_i(r_j, \sigma_j)$ is a single-particle "spin-orbital" comprising a spatial function, $\psi_i(r_j)$, and a spin function, $\alpha_i(\sigma_j)$. It's important to note that $\psi_i(r_j)$ is spin-independent in closed-shell systems, aligning with the "spin-restricted Hartree–Fock approximation." These spin-orbitals must not only be linearly independent but also orthonormal to simplify the equations significantly. It can be straightforwardly demonstrated that $\Phi$ is normalized to 1.

Moreover, if the Hamiltonian does not depend on spin or is diagonal in the spin basis $\sigma = |\uparrow\rangle; |\downarrow\rangle$, the expected value of the Hamiltonian using Hartree atomic units with the wavefunction $\Phi$ is calculated as follows:

$$
\begin{aligned}
\langle\Phi|\hat{H}|\Phi\rangle = &\sum_{i,\sigma}\int dr\psi_i^{\sigma*}(r)\left(-\frac{1}{2}\nabla^2 + V_{\text{ext}}(r)\right)\psi_i^\sigma(r) + E_{II} \\
&+ \frac{1}{2}\sum_{i,j,\sigma_i,\sigma_j}\int drdr'\psi_i^{\sigma_i*}(r)\psi_j^{\sigma_j*}(r')\frac{1}{|r-r'|}\psi_i^{\sigma_i}(r)\psi_j^{\sigma_j}(r') \\
&- \frac{1}{2}\sum_{i,j,\sigma}\int drdr'\psi_i^{\sigma*}(r)\psi_j^{\sigma*}(r')\frac{1}{|r-r'|}\psi_j^\sigma(r)\psi_i^\sigma(r').
\end{aligned}
\tag{28}
$$

The first term in the Hartree–Fock method consolidates the single-body expectation values by summing over the orbitals, while the third and fourth terms address the direct and exchange interactions between electrons, involving double sums. Conventionally, we include the "self-interaction" (where $i = j$), a concept deemed spurious but necessary as it cancels out in the aggregation of direct and exchange interactions. When this self-interaction is accounted for, the cumulative effect across all orbitals represents the electron density, and the direct term effectively simplifies to the Hartree energy. The "exchange" term specifically interacts between electrons of the same spin, owing to the orthogonality of spin components in orbitals for opposite spins. These interactions, crucial to the energy computation. The essence of the Hartree–Fock approach is to minimize the total energy subject to all constraints imposed by the wavefunction's degrees of freedom. This minimization process leverages orthonormality to streamline the equations and employs Lagrange multipliers to ensure this condition is met throughout.

## A.2 Real spherical harmonics functions

A real basis of spherical harmonics $Y_{lm} : S^2 \to \mathbb{R}$ can be defined in terms of their complex analogues $Y_l^m : S^2 \to \mathbb{C}$ by setting

$$
Y_{lm} = \begin{cases}
\frac{i}{\sqrt{2}}\left(Y_l^{-|m|} - (-1)^m Y_l^{|m|}\right) & \text{if } m < 0 \\
Y_l^0 & \text{if } m = 0 \\
\frac{1}{\sqrt{2}}\left(Y_l^{-|m|} + (-1)^m Y_l^{|m|}\right) & \text{if } m > 0
\end{cases}
\tag{29}
$$

The Condon–Shortley phase convention is frequently employed to maintain uniformity and consistency across the analysis. The corresponding inverse equations defining the complex spherical harmonics $Y_l^m : S^2 \to \mathbb{C}$ in terms of the real spherical harmonics $Y_{lm} : S^2 \to \mathbb{R}$ are

$$
Y_l^m = \begin{cases}
\frac{1}{\sqrt{2}}\left(Y_{l|m|} - iY_{l,-|m|}\right) & \text{if } m < 0 \\
Y_{l0} & \text{if } m = 0 \\
\frac{(-1)^m}{\sqrt{w}}\left(Y_{l|m|} + iY_{l,-|m|}\right) & \text{if } m > 0
\end{cases}
\tag{30}
$$

It is well-established from the analytical solutions of the hydrogen atom that the eigenfunctions corresponding to the angular component of the wave function manifest as spherical harmonics. Intriguingly, when magnetic terms are absent, the solutions to the non-relativistic Schrödinger equation can be rendered as real functions. This characteristic underpins the prevalent use of real-form basis functions in electronic structure computations, as it obviates the need for complex algebra in the software implementations. It is essential to recognize that these real-valued functions occupy an identical functional space to that of their complex counterparts, ensuring no loss of generality or completeness in the solutions.

## A.3 Irreps representation and tensor product

**Irreps representation** The model's equivariant framework harnesses the special orthogonal group SO(3) to represent the 3D rotational symmetries that are fundamental to molecular structures. It utilizes the irreducible representations (irreps) of SO(3), denoted by an integer $l$, which resonate with spherical harmonic functions $Y_{lm}$. These harmonics confer rotational attributes upon the feature vectors, granting the model rotational equivariance and allowing for uniform geometric property assessment.

**Tensor product**  In pursuit of heightened expressiveness within the model, it orchestrates interactions among irrep features corresponding to different angular momenta $l$ via the tensor product. This operation combines two irreps with angular momenta $l_1$ and $l_2$ to form a new irrep characterized by angular momentum $l_3$, employing Clebsch-Gordan coefficients to facilitate an expansion that is weighted by $w_{m_1,m_2}$:

$$
\begin{aligned}
h_{l_3,m_3} &= h_{l_1,m_1} \otimes h_{l_2,m_2} \\
&= \sum_{m_1,m_2} w_{m_1,m_2} C^{l_3,m_3}_{l_1,m_1,l_2,m_2} h_{l_1,m_1} h_{l_2,m_2}.
\end{aligned}
\tag{31}
$$

This mathematical formulation enables the synthesis of complex features from simpler ones, capturing the intricate interplay of angular momentum in the molecular description.

## A.4  Wigner–Eckart theorem

The Wigner-Eckart theorem [46], a cornerstone of representation theory, articulates that the matrix elements of spherical tensor operators, when projected onto a basis of angular momentum eigenstates, can be factorized into two distinct components: a term invariant to angular momentum orientation and a Clebsch-Gordan coefficient. It provides a bridge connecting the symmetry transformation groups governing spatial configurations, as applied within the Schrödinger equation framework, to the fundamental conservation principles of energy, momentum, and angular momentum.

Mathematically articulated, the Wigner-Eckart theorem posits that for a given tensor operator $T^{(k)}$ and two states with angular momenta $j$ and $j'$, there exists a reduced matrix element $\langle j|T^{(k)} vert j'\rangle$ that is invariant with respect to the magnetic quantum numbers $m$, $m'$, and $q$. The theorem asserts that for all values of these quantum numbers, the matrix elements of the tensor operator satisfy the relation:

$$
\langle jm|T_q^{(k)}|j'm'\rangle = \langle j'm'kq|jm\rangle\langle j \parallel T^{(k)} \parallel j'\rangle,
\tag{32}
$$

where $T_q^{(k)}$ is the $q$-th component of the spherical tensor operator $T^{(k)}$ of rank $k$, and $|jm\rangle$ represents an eigenstate of the total angular momentum operator $J^2$ and its $z$-component $J_z$. $\langle j'm'kq|jm\rangle$ is the Clebsch-Gordan coefficient, which mediates the coupling of angular momentum $j'$ with $k$ to yield $j$. $\langle j \parallel T^{(k)} \parallel j'\rangle$ symbolizes the reduced matrix element, a scalar that encapsulates the essence of the tensor operator independent of the magnetic quantum numbers. This theorem elegantly decouples the geometric dependencies from the dynamic properties of the matrix elements, thereby simplifying the computation of transition amplitudes.

## A.5  Proof of the proportionality between on-site three-index overlap integrals and Clebsch-Gordan coefficients

The order-$l$ equivariant embeddings $h_A^l$ using on-site three-index overlap integrals $\tilde{Q}$:

$$
\begin{aligned}
\tilde{Q}^{\mu,\nu}_{nlm} &= \tilde{Q}^{n_1,l_1,m_1;n_2,l_2,m_2}_{nlm} \\
&= \int_{r\in\mathbb{R}^3} \left(\Phi_A^{n_1,l_1,m_1}(r)\right)^* \Phi_A^{n_2,l_2,m_2}(r)\tilde{\Phi}_A^{n,l,m}(r)dr \\
&= \int_0^\infty \int_0^{2\pi} \int_0^\pi c_{n_1,l_1} \exp\left(-\gamma_{n_1,l_1}r^2\right) r^{l_1} Y_{l_1}^{m_1*}(\theta,\varphi) \\
&\qquad\qquad c_{n_2,l_2} \exp\left(-\gamma_{n_2,l_2}r^2\right) r^{l_2} Y_{l_2}^{m_2}(\theta,\varphi) \\
&\qquad\qquad c_{n,l} \exp\left(-\gamma_{n,l}r^2\right) r^l Y_l^m(\theta,\varphi)r^2 \sin\theta dr d\theta d\phi \\
&= \int_0^\infty c_{n_1,l_1} \exp\left(-\gamma_{n_1,l_1}r^2\right) r^{l_1} c_{n_2,l_2} \exp\left(-\gamma_{n_2,l_2}r^2\right) r^{l_2} c_{n,l} \exp\left(-\gamma_{n,l}r^2\right) r^l r^2 dr \\
&\qquad \int_0^{2\pi} \int_0^\pi Y_{l_1}^{m_1*}(\theta,\varphi)Y_{l_2}^{m_2}(\theta,\varphi)Y_l^m(\theta,\varphi) \sin\theta d\theta d\phi \\
&\propto \int d\Omega Y_{l_1}^{m_1*}(\hat{r})Y_{l_2}^{m_2}(\hat{r})Y_l^m(\hat{r}) \\
&= \sqrt{\frac{(2l_1+1)(2l_2+1)}{4\pi(2l+1)}} \begin{pmatrix} l_1 & l_2 & l \\ 0 & 0 & 0 \end{pmatrix} \begin{pmatrix} l_1 & l_2 & l \\ m_1 & m_2 & m \end{pmatrix},
\end{aligned}
\tag{33}
$$

where $\begin{pmatrix} l_1 & l_2 & l \\ m_1 & m_2 & m \end{pmatrix}$ denotes the Wigner $3j$-symbols, which are proportional to the Clebsch-Gordan coefficients.

## A.6 Auxiliary Gaussian-type basis functions

Adhering to the methodology outlined in Ref. [36], we define the auxiliary Gaussian-type basis functions, and for the sake of brevity, we consider their form at the origin, $\mathbf{x}_A = 0$:

$$\tilde{\Phi}^{n,l,m}(\mathbf{r}) := c_{n,l} \cdot \exp(-\gamma_{n,l} \cdot r^2) \, r^l \, Y_{lm}(\frac{\mathbf{r}}{r}) \tag{34}$$

where $c_{n,l}$ is a normalization constant such that $\int_{\mathbf{r}} ||\tilde{\Phi}_A^{n,l,m}(\mathbf{r}))||^2 d\mathbf{r} = 1$ following standard conventions. For numerical experiments considered in this work the scale parameters $\gamma$ are chosen as (in atomic units):

$$\gamma_{n,l=0} := 128 \cdot (0.5)^{n-1} \quad \text{where} \quad n \in \{1, 2, \cdots, 16\} \tag{35}$$

$$\gamma_{n,l=1} := 32 \cdot (0.25)^{n-1} \quad \text{where} \quad n \in \{1, 2, \cdots, 8\} \tag{36}$$

$$\gamma_{n,l=2} := 4.0 \cdot (0.25)^{n-1} \quad \text{where} \quad n \in \{1, 2, 3, 4\} \tag{37}$$

Note that the auxiliary basis $\tilde{\Phi}_A$ is independent of the atomic numbers thus the resulting $\mathbf{h}_A$ are of equal length for all chemical elements.

## A.7 Deep equilibrium model

Here, we adopt a generalized notation for the DEQ [31] function, denoted as $f(z, x)$. This notation encapsulates the earlier specified function $f(x, y) = \sigma(Wz + Ux + b)$, where $\sigma$ represents the activation function, and $W$, $U$, and $b$ denote the weight matrix, input weight matrix, and bias vector, respectively. Our objective here is to determine a stable fixed point $z^*$ that satisfies the equilibrium condition $z^* = f(z^*, x)$.

Any deep network—irrespective of its depth or the intricacy of its connections—can be succinctly encapsulated within the framework of a single-layer DEQ model. Remarkably, this encapsulation does not succumb to the typical exponential surge in parameters that is often associated with universal function approximation theorems for single-layer models. In essence, a single-layer DEQ model can represent the functional capacity of any network without necessitating an increase in the number of parameters. Consider a conventional deep network described by the composition of two functions, denoted as $y = g_2(g_1(x))$. This construct can be seamlessly transmuted into a single-layer DEQ model by amalgamating all intermediate variables derived from the computation into an extended vector,

$$f(z, x) = f\left(\begin{bmatrix} z_1 \\ z_2 \end{bmatrix}, x\right) = \begin{bmatrix} g_1(x) \\ g_2(z_1) \end{bmatrix}. \tag{38}$$

At an equilibrium point $z^*$ of this function,

$$z^* = f(z^*, x) \iff z_1^* = g_1(x), \quad z_2^* = g_2(z_1^*) = g_2(g_1(x)). \tag{39}$$

We can concatenate all intermediate results of a computational graph into the vector $z$, and designate the function $f$ as the operator that applies the "subsequent" computation in the graph to each of these elements. This theoretical framework unequivocally demonstrates the representational strength of a single DEQ layer.

To elucidate the principles of implicit backpropagation tailored to DEQ models—and by extension, to any fixed-point iterative layer—we will primarily concentrate on the DEQ model's specific form herein. Our objective is to compute the vector-Jacobian product $(\frac{\partial z^*(\cdot)}{\partial(\cdot)})^T y$ for a given vector $y$, where $(\cdot)$ symbolizes any variable with respect to which we seek to differentiate the fixed point, all of which influence the determined fixed point $z^*$. Differentiating both sides of the fixed-point equation yields:

$$\frac{\partial z^*(\cdot)}{\partial(\cdot)} = \frac{\partial f(z^*(\cdot), x)}{\partial(\cdot)} = \frac{\partial f(z^*, x)}{\partial z^*} \frac{\partial z^*(\cdot)}{\partial(\cdot)} + \frac{\partial f(z^*, x)}{\partial(\cdot)}, \tag{40}$$

where $z^*(\cdot)$ denotes the case where $z^*$ is treated as an implicit function of the differentiated quantity, and $z^*$ alone signifies the equilibrium value. The second equality arises by applying the multivariate chain rule. Rearranging terms, we arrive at an explicit expression for the Jacobian:

$$\frac{\partial z^*(\cdot)}{\partial(\cdot)} = (I - \frac{\partial f(z^*, x)}{\partial z^*})^{-1} \frac{\partial f(z^*, x)}{\partial(\cdot)}, \tag{41}$$

where the terms on the right-hand side can be computed using conventional automatic differentiation. To compute the vector-Jacobian product,

$$\left(\frac{\partial z^*(\cdot)}{\partial(\cdot)}\right)^T y = \left(\frac{\partial f(z^*, x)}{\partial(\cdot)}\right)^T \left(I - \frac{\partial f(z^*, x)}{\partial z^*}\right)^{-T} y. \tag{42}$$

In practice, the focal term is the solution to the linear system (abbreviated as $g$):

$$g = \left(I - \frac{\partial f(z^*, x)}{\partial z^*}\right)^{-T} y, \tag{43}$$

which can be expressed as a fixed-point equation:

$$g = \left(\frac{\partial f(z^*, x)}{\partial z^*}\right)^T g + y. \tag{44}$$

The convergence of this equation via forward iteration hinges on the stability of the Jacobian $\frac{\partial f(z^*, x)}{\partial z^*}$, which also underlies the local stability of the forward iterative process.

The derivation of the vector-Jacobian product for a DEQ layer can be distilled into a two-step procedure:

- Solve the Fixed-Point Equation $g = \left(\frac{\partial f(z^*, x)}{\partial z^*}\right)^T g + y$. This can be done either through direct analytical inversion or, more commonly, by employing an iterative method that necessitates only the multiplications by $\left(\frac{\partial f(z^*, x)}{\partial z^*}\right)^T$. Such multiplications can be expediently performed using standard automatic differentiation, as they are vector-Jacobian products in their own right.

- Calculate the final vector-Jacobian product $\left(\frac{\partial z^*(\cdot)}{\partial(\cdot)}\right)^T y = \left(\frac{\partial f(z^*, x)}{\partial(\cdot)}\right)^T g$. This step also leverages the capabilities of conventional automatic differentiation, as it involves the computation of another vector-Jacobian product.

By executing these steps, one can integrate DEQ layers into the backpropagation algorithm, thereby harnessing the power of implicit differentiation to effectively train deep equilibrium models.

## A.8 Metrics

To evaluate the accuracy and computational efficiency of the predicted Hamiltonian matrix, we adopt a set of metrics as demonstrated in Ref. [32] to evaluate the model's performance:

**MAE of Hamiltonian matrix** $H$ The MAE against DFT-computed reference data, considering both diagonal and off-diagonal blocks that represent intra- and inter-atomic interactions, respectively. Given the sparsity induced by distant atom pairs in larger molecules, we separately assess the MAE for diagonal and off-diagonal blocks along with the total MAE of the matrix.

**MAE of occupied orbital energies** $\varepsilon$ The MAE for occupied orbital energies, including HOMO and LUMO levels derived from the predicted and reference Hamiltonian matrices, is used to measure the matrix's predictive accuracy for these critical properties.

**Cosine similarity of orbital coefficients** $\psi$ To determine the similarity between predicted and reference electronic wavefunctions, we compute the cosine similarity of the coefficients for occupied molecular orbitals, which are crucial for inferring chemical properties.

Table 3: The performance of DFT calculation acceleration.

| Training Dataset | DFT initialization | Metric | QHNet | DEQHNet |
|---|---|---|---|---|
| QH9-stable-id | 1e | Optimal ratio | $0.057 \pm 0.004$ | $0.060 \pm 0.004$ |
| | | Achieved ratio ↓ | $0.395 \pm 0.030$ | $\mathbf{0.363 \pm 0.074}$ |
| | | Error-level ratio ↑ | $0.635 \pm 0.039$ | $\mathbf{0.683 \pm 0.039}$ |
| | minao | Optimal ratio | $0.102 \pm 0.005$ | $0.113 \pm 0.006$ |
| | | Achieved ratio ↓ | $0.706 \pm 0.031$ | $\mathbf{0.681 \pm 0.129}$ |
| | | Error-level ratio ↑ | $0.408 \pm 0.025$ | $\mathbf{0.459 \pm 0.031}$ |
| QH9-stable-ood | 1e | Optimal ratio | $0.057 \pm 0.004$ | $0.060 \pm 0.004$ |
| | | Achieved ratio ↓ | $0.400 \pm 0.030$ | $\mathbf{0.359 \pm 0.031}$ |
| | | Error-level ratio ↑ | $0.620 \pm 0.037$ | $\mathbf{0.678 \pm 0.038}$ |
| | minao | Optimal ratio | $0.102 \pm 0.005$ | $0.113 \pm 0.006$ |
| | | Achieved ratio ↓ | $0.715 \pm 0.033$ | $\mathbf{0.674 \pm 0.045}$ |
| | | Error-level ratio ↑ | $0.406 \pm 0.021$ | $\mathbf{0.454 \pm 0.029}$ |
| QH9-dynamic-100k-geo | 1e | Optimal ratio | $0.056 \pm 0.006$ | $0.056 \pm 0.006$ |
| | | Achieved ratio ↓ | $0.392 \pm 0.036$ | $\mathbf{0.373 \pm 0.037}$ |
| | | Error-level ratio ↑ | $0.648 \pm 0.041$ | $\mathbf{0.673 \pm 0.037}$ |
| | minao | Optimal ratio | $0.098 \pm 0.008$ | $0.098 \pm 0.008$ |
| | | Achieved ratio ↓ | $0.679 \pm 0.041$ | $\mathbf{0.646 \pm 0.043}$ |
| | | Error-level ratio ↑ | $0.443 \pm 0.044$ | $\mathbf{0.475 \pm 0.050}$ |
| QH9-dynamic-100k-mol | 1e | Optimal ratio | $0.056 \pm 0.006$ | $0.056 \pm 0.006$ |
| | | Achieved ratio ↓ | $0.512 \pm 0.138$ | $\mathbf{0.426 \pm 0.081}$ |
| | | Error-level ratio ↑ | $0.622 \pm 0.048$ | $\mathbf{0.644 \pm 0.048}$ |
| | minao | Optimal ratio | $0.098 \pm 0.008$ | $0.098 \pm 0.008$ |
| | | Achieved ratio ↓ | $0.882 \pm 0.217$ | $\mathbf{0.736 \pm 0.128}$ |
| | | Error-level ratio ↑ | $0.406 \pm 0.066$ | $\mathbf{0.436 \pm 0.050}$ |

**Acceleration ratio**   Acceleration ratios, such as the achieved ratio and the error-level ratio, could be used to evaluate how well the predicted Hamiltonian matrix speeds up DFT calculations. These ratios compare the number of optimization steps needed when using the predicted matrix versus traditional initial guesses, and the steps required to reach a comparable error level in the DFT SCF cycle.

### A.9   Accelerate ratio of DEQHNet

The DEQHNet, after being trained on the QH9 dataset, undergoes evaluation on a meticulously selected subset of molecules. This subset consists of 50 molecules that have been randomly chosen from the overlapping elements of the QH9-stable and QH9-dynamic's respective test subsets. The intersection ensures that the evaluation is conducted on a common ground, facilitating a direct comparison of the performance results between the stable and dynamic regimes. As illustrated in Table 3, the study underscores the feasibility of leveraging DEQHNet as initial seeds for electronic structure computations, which can potentially reduce the time to convergence significantly.

### A.10   Explanation of the error in orbital energy

In the experiments with MD17 and QH9, we observed a decrease in the Mean Absolute Error (MAE) of the Hamiltonian, but the MAE of the orbital energy did not show a corresponding positive trend. To investigate this, we conducted additional experiments. We randomly selected a molecule, added a Hermitian Gaussian noise matrix to its Hamiltonian, and solved the corresponding generalized eigenvalue equation. As shown in Fig. 6, it can be seen that as the noise on the Hamiltonian gradually increases, the range of error in the orbital energy becomes larger. The errors in the orbital energy of QHNet and DEQHNet on MD17 and QH9 are all within the range shown in the figure. Only when the MAE of the Hamiltonian is sufficiently small can it be ensured that the corresponding orbital energy error is also small. In the comparison between QHNet and PhiSNet, we observed similar situations. This somewhat suggests that using existing Hamiltonian error measurements cannot definitively reflect the errors in the eigenvalues. These evidence suggests that it is a nontrivial and long-standing challenge in the domain of Hamiltonian prediction to design a proper metric on the matrix space so that reflects the metric of derived quantities e.g. energy, for which we will investigate in future work.

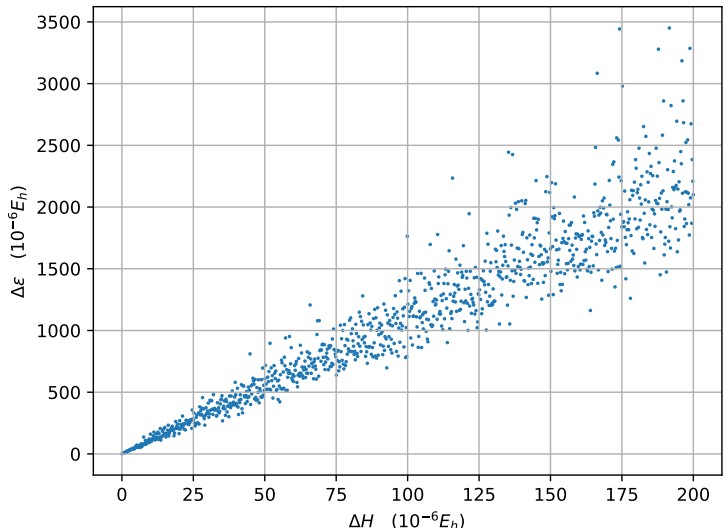

Figure 6: The variation in the error of orbital energy with the injection of Hermitian Gaussian noise to the Hamiltonian.

Table 4: Hyperparameters of DEQHNet for different datasets.

| | | MD17 | | | QH9 | | | |
| | Water | Ethanol | Malonaldehyde | Uracil | Stable-id | Stable-ood | Dynamic-geo | Dynamic-mol |
|---|---|---|---|---|---|---|---|---|
| # Train/validation/test | 500/500/4,000 | 25,000/500/4,500 | 25,000/500/1,478 | 25,000/500/4,500 | 104,664/13,083/13,084 | 104,001/17,495/9,335 | 79,920/9,990/9,990 | 79,900/9,900/10,100 |
| Cutoff (Å) | 15.0 | 15.0 | 15.0 | 15.0 | 15.0 | 15.0 | 15.0 | 15.0 |
| Order of spherical harmonics | 4 | 4 | 4 | 4 | 4 | 4 | 4 | 4 |
| # layers | 5 | 5 | 5 | 5 | 5 | 5 | 5 | 5 |
| # neurons | 128 | 128 | 128 | 128 | 128 | 128 | 128 | 128 |
| Batch size | 10 | 5 | 5 | 5 | 32 | 32 | 32 | 32 |
| Learning rate (LR) | 5e-4 | 1e-3 | 1e-3 | 1e-3 | 5e-4 | 5e-4 | 5e-4 | 5e-4 |
| Optimizer | AdamW | AdamW | AdamW | AdamW | AdamW | AdamW | AdamW | AdamW |
| EMA | True | True | True | True | False | False | False | False |
| EMA start epoch | 40 | 0 | 0 | 0 | NA | NA | NA | NA |
| # iterations (Forward) | 40 | 40 | 40 | 40 | 3 | 3 | 3 | 3 |
| # iterations (Backward) | 40 | 40 | 40 | 40 | 3 | 3 | 3 | 3 |

## A.11 Settings of experiments

The overarching framework is implemented using PyTorch [47], PyTorch Geometric [48], e3nn [49], and Torchdeq [50] libraries. The computation of on-site three-index overlap integrals is conducted by Psi4 software [51], and subsequently converted to adhere to the conventions of PySCF [41]. The loss function for training DEQHNet model is the summation of the Frobenius norm and L1 norm of the absolute error of the matrix of Hamiltonians,

$$L = \sqrt{\frac{1}{N^2} \sum_{i,j} (H_{i,j} - \hat{H}_{i,j})^2} + \frac{1}{N^2} \sum_{i,j} \left| H_{i,j} - \hat{H}_{i,j} \right|, \tag{45}$$

where $N$ is the number of elements in Hamiltonian matrix. $H$ and $\hat{H}$ denote the predicted and ground-truth Hamiltonian respectively. AdamW optimizer [52] was adopted. DEQHNet model was trained on single 16G Nvidia Tesla V100 GPU for MD17 dataset and single 80G Nvidia Tesla A100 GPU for QH9 dataset. It is noteworthy that the iterative process inherent in DEQ may impact training efficiency to a certain extent. In our experiments with the MD17 dataset, we observed that the model eventually converges after only 2-3 DEQ iterations. Therefore, to enhance training efficiency, we set the maximum number of iterations for both the forward and backward passes of DEQ to 3 in our experiments with the QH9 dataset. With this configuration, the training time on the QH9 data was approximately 1.5 times that of QHNet. Furthermore, detailed settings of hyperparameters are summarized in the Table 4.

## A.12 Rethinking and Prospects

The Hamiltonian exhibits self-consistent iterative properties. Specifically, in the DFT solution process, the information from the previously obtained Hamiltonian is used to construct the current Hamiltonian

for the current DFT solution. This process is repeated until certain criteria meet a set threshold. We hypothesize that introducing DEQ enables the network to learn the iterative mechanism of solving the Hamiltonian, rather than directly learning the solution to the Hamiltonian. This shift could allow the network to better capture inherent physical laws, thereby improving generalizability. Compared to models without self-consistency, DEQ introduces additional orbital information to the network input. This, coupled with the introduction of scientific priors, can expedite DFT convergence even when the model's output is simply used as the initial value for DFT. In contrast with methods that introduce self-consistency (which often include DFT computations in the loss function during model training), our method offers an architectural enhancement. The DEQH model does not directly include DFT computations, leading to a more efficient computational process. Empirically, if the dataset and molecular size are relatively small and the cost of DFT computations during training is acceptable, previous methods introducing self-consistency might outperform ours (as suggested by DEQHNet's results on water, where DEQHNet requires more data). However, when the dataset is large enough, or when it includes large molecules where DFT computation cost is prohibitive, the benefits of the DEQH model become more significant.

The DEQH model is a versatile approach that can be seamlessly integrated with current machine learning models for Hamiltonian prediction. In this paper, we opted for QHNet [26] over PhiSNet [23] due to the current implementation of PhiSNet being restricted to single-molecule support. This limitation stems from PhiSNet's matrix prediction module, which is tailored to predict matrices of fixed size for identical molecules [32]. Additionally, based on the ablation studies, we also posit that providing the overlap matrix could enable training on data whose DFT labels are associated with varying basis sets, whereas models lacking it may be restricted to consistent DFT-level datasets. Furthermore, the DEQH model employs a method that is entirely orthogonal to the inclusion of iterative DFT processes during training to achieve the self-consistency of the Hamiltonian—an architectural innovation within the model. Incorporating DFT iterations into the existing Hamiltonian training introduces a more explicit physical prior [? 28, 29, 30] compared to the DEQH model, which requires a larger dataset to enable the model to learn the iterative solving process. However, for cases involving large datasets and larger molecules, the DEQH model may be more appropriate. This not only offers the possibility of direct integration with off-the-shelf training methodologies that incorporate self-consistency but may also assist in enhancing aspects such as model generalizability.

