# OpenReview forum: "Infusing Self-Consistency into Density Functional Theory Hamiltonian Prediction via Deep Equilibrium Models"
_NeurIPS.cc/2024/Conference — NeurIPS 2024 poster_

### Official Review · Reviewer_mPEr · 2024-07-06

**Soundness:** 2
**Presentation:** 2
**Contribution:** 3
**Rating:** 5
**Confidence:** 2

**Summary:**

Paper presents DEQH model which is a Deep Equilibrium Model to solve quantum Hamiltonians. DEQH uses the deep equilibrium model because it converges to a fixed point, which matches the self-consistent nature of Hamiltonian solving. Paper presents architecture based on the QHNet, results and comparison with QHNet for the MD17 and QH9 datasets, and convergence and ablation study.

**Strengths:**

Paper addresses a significant problem of computational chemistry. Paper is original and novel in integrating deep equilibrium model with Hamiltonian solving (combining fixed point convergence and self-consistency property makes sense). Paper has good set of experiments comparing with an alternative state of the art model and reaches better results than other model.

**Weaknesses:**

It seems there are differences between DEQH and QHNet, including the input of the predicted H, but the differences are not clearly highlighted in the paper. this makes the comparison harder to interpret.

The result of H, Psi errors being low while epsilon errors are high for DEQH is strange. What is the meaning of H, epsilon, and Psi, and why is the epsilon error higher in many cases? No error bars on tables and ablation study, makes results harder to interpret.

**Questions:**

see above.

**Limitations:**

A limitation I see is that DFT ground truth calculation is still used in the loss. Since DEQH model is a Hamiltonian solver, it would be nice to see how it performs using a self-consistency loss. It seems that avoiding having to run many DFT calculations would be a big improvement, whereas the current results of paper show DEQH gets lower error results than previous model but still needs DFT data and requires 1.5x training time.

---

> ### Author Rebuttal · Authors · 2024-08-05
>
> Thanks for your review. We will address each point in our response accordingly.
>
> ## Weakness 1:
> Thank you for your feedback. DEQH is a general method specifically devised to instill self-consistency into pre-existing models, while QHNet is a distinct network that predicts the Hamiltonian directly. Specifically, DEQH functions as a Hamiltonian solver, learning the iterative solving process $H^* = f(H^*, Z, R)$, while QHNet operates as a predictor $H = f(Z, R)$, where $H, Z, R$ denote Hamiltonian matrix, atomic number, and atomic coordinates resepctively. f represents a neural network and $H^*$ denotes the fixed point of the equation. In our research, we utilized QHNet as the fundamental backbone and incorporated DEQH into it, thereby creating DEQHNet. As mentioned in the Supplementary Material A.11, we opted for QHNet over PhiSNet due to the current implementation of PhiSNet being restricted to single-molecule support. This limitation stems from PhiSNet’s matrix prediction module, which is tailored to predict matrices of fixed size for identical molecules.
>
> DEQH introduces self-consistency by approaching the problem of Hamiltonian solution from the vantage point of a fixed-point iteration, which necessitates the inclusion of the Hamiltonian in the network's input. To ensure the broad applicability of the Hamiltonian across various graph neural networks, it must be transformed into signals on the graph. This is demonstrated in Section 4.1, where we transposed the Hamiltonian, overlap matrix, and other equivalent tensors into invariant node features.
>
> Furthermore, we provide a PDF document in the global rebuttal section, which includes a figure delineating the distinction between the off-the-shelf model used for Hamiltonian prediction and the DEQH model. We understand the importance of clearly communicating these differences and will further elaborate on these points in the revised version of our paper.
>
> ## Weakness 2:
> Thank you for your question. In Appendix A.7, we provide detailed definitions of the metrics. H refers to the Mean Absolute Error (MAE) of the Hamiltonian matrix, while epsilon and Psi are the eigenvalues (orbital energy) and eigenvectors (orbital coefficients) obtained by solving generalized eigenvalue problems. The metrics for epsilon and Psi are the MAE of the orbital energies and the cosine similarity of the orbital coefficients, respectively.
>
> We explain in Appendix A.9 why epsilon is higher than the baseline in many cases. We selected a molecule randomly, added a Hermitian-Gaussian noise matrix to its Hamiltonian, and solved the corresponding generalized eigenvalue equation. We observed that as the noise on the Hamiltonian increased, the range of errors in the orbital energy also increased. The orbital energy errors reported for both QHNet and DEQHNet on the MD17 and QH9 datasets fall within this range, suggesting the experimental results are reasonable. This implies that only when the MAE of the Hamiltonian is sufficiently small can we ensure the corresponding orbital energy deviation is also small. In repeated experiments on the MD17 dataset, we found that when the Hamiltonian MAE of the model is at the same level, it is possible to obtain both a lower and a higher orbital energy MAE than the baseline. We believe this contributes to the performance difference observed between DEQHNet and the baseline model. This somewhat suggests that using existing Hamiltonian error measurements cannot definitively reflect the errors in the eigenvalues. This is a nontrivial, separate research question that we plan to investigate in the future.
>
> We will elaborate on this point more clearly in the revised version of our paper to provide a comprehensive understanding of the model's performance.
>
> ## Limitations:
> Thank you for your insightful comments. Indeed, the DEQH model introduces self-consistency using DEQ, but it remains a supervised learning task that requires DFT data as labels. The statement "DEQH model is a Hamiltonian solver" refers to the DEQH model's ability to learn the iterative process of solving the Hamiltonian through DFT data, rather than it being a Partial Differential Equation (Schr\"odinger equation) solver. With this mechanism, DEQH is a better way for supervised learning of Hamiltonian in that it exploits more information in a model of limited size by passing through the model multiple times until convergence, and that the model learns the iteration map which is a physical mechanism that generalizes better.
>
> Your idea of bypassing DFT computations is intriguing and aligns with our thoughts. As we mentioned in Appendix A.11, some existing methods introduce self-consistency using unlabeled data. Our approach is entirely orthogonal to these methods. Therefore, combining these methods to achieve a trade-off between DFT computational cost and accuracy is a promising direction worth exploring.

---

> ### Comment · Reviewer_mPEr · 2024-08-12
>
> Thank the authors for response. I still think the ablation study is hard to interpret, but I appreciate the additional comparison with other benchmarks. I have increased score accordingly

---

> ### Author Response · Authors · 2024-08-13
>
> We are pleased that our response has satisfied you and are very grateful for the increased score.
>
> As outlined in our rebuttal, the DEQH functions as a Hamiltonian solver, learning the iterative solving process $H^* = f(H^*, Z, R)$, while the QHNet acts as a predictor with the formula $H = f(Z, R)$. In practice, if the dataset includes overlap matrices, the network input can also incorporate this information, treating it similarly to the Hamiltonian by transforming it into equivariant node features (even if it's absent from the dataset, the calculation of the overlap matrix is quite straightforward and can be generated during data preprocessing). The overlap matrix offers a wealth of detailed information and can be computed easily with $Z$ and $R$. Accordingly, the equations for the DEQH and the modified Hamiltonian predictor become $H^* = f(H^*, Z, R, S)$ and $H = f(Z, R, S)$, respectively, where $S$ is the overlap matrix. In line with this, we conducted several experiments in our ablation study, testing the QHNet ($H = f(Z, R)$), QHNet w/ S ($H = f(Z, R, S)$), DEQHNet ($H^* = f(H^*, Z, R, S)$), and DEQHNet w/o S ($H^* = f(H^*, Z, R)$) on the Uracil dataset. The results depicted in Figure 4 indicate that:
> * Incorporating the overlap matrix as an input leads to a lower Mean Absolute Error (MAE) in predicting the Hamiltonian (DEQHNet's Hamiltonian MAE is lower than DEQHNet w/o S, and QHNet's Hamiltonian MAE is higher than QHNet w/ S). Furthermore, the inclusion of the overlap matrix results in higher similarity in the orbital coefficients, suggesting that the overlap matrix is beneficial for the network's learning of the Hamiltonian and also supports our experimental hypothesis that the overlap matrix provides orbital information, which in turn improves the results for the orbital coefficients.
> * Regardless of the presence of the overlap matrix in the network's input, the DEQH's Hamiltonian MAE is consistently lower than that of the QHNet-based Hamiltonian MAE, indicating that the DEQH model benefits from the introduction of self-consistency.
>
> We will ensure that all new experimental results are included in the manuscript and will promptly refine our paper as soon as we are permitted a polish for the final version. We deeply appreciate your valuable feedback.

---

### Official Review · Reviewer_DtX8 · 2024-07-12

**Soundness:** 3
**Presentation:** 3
**Contribution:** 3
**Rating:** 7
**Confidence:** 2

**Summary:**

The paper introduces a novel neural network architecture DEQH, extending deep equilibrium models (DEQs) to improve predictions of quantum Hamiltonians. The architecture constrains solutions to ensure self-consistency of the Hamiltonian, thereby improve generalization capability and test accuracy.

**Strengths:**

Constraining the network to obey self-consistency is natural thing to do and matches the physical constraints of the problem at hand, without the need for costly DFT calculations.

The paper proposes an elegant parameterization. Further, it provides an extensive empirical analysis showing the practical benefits of the proposed approach on several datasets.

**Weaknesses:**

As mentioned, DEQH model acts fundamentally as a solver, iteratively determining the Hamiltonian with fixed point iterations. It is not directly clear to me why this would necessarily be better, or how this compares, to methods that integrating frameworks that also ensure self-consistency. Why would we expect to be better both computationally and in terms of performance, or is there a clear trade-off?

**Questions:**

How do we expect DEQH to compare to methods that ensure self-consistency through integration? Is there are trade-off between performance and computational cost, or do we expect the method to be better overall. Why?

**Limitations:**

The paper proposes a novel way to embed self-consistency into neural network architectures for quantum Hamiltonian prediction and provides empirical evidence of benefits.

---

> ### Author Rebuttal · Authors · 2024-08-05
>
> We sincerely appreciate the reviewer's recognition of the innovation and significance of our work and will address each point in our response accordingly.
>
> ## Weakness:
> Thank you for your insightful question. The Hamiltonian exhibits self-consistent iterative properties. Specifically, in the DFT solution process, the information from the previously obtained Hamiltonian is used to construct the current Hamiltonian for the current DFT solution. This process is repeated until certain criteria meet a set threshold. We hypothesize that introducing DEQ enables the network to learn the iterative mechanism of solving the Hamiltonian, rather than directly learning the solution to the Hamiltonian. This shift could allow the network to better capture inherent physical laws, thereby improving generalizability. Compared to models without self-consistency, DEQ introduces additional orbital information to the network input. This, coupled with the introduction of scientific priors, can expedite DFT convergence even when the model's output is simply used as the initial value for DFT (as shown in Appendix A.8). In contrast with methods that introduce self-consistency (which often include DFT computations in the loss function during model training), our method offers an architectural enhancement. The DEQH model does not directly include DFT computations, leading to a more efficient computational process. Empirically, if the dataset and molecular size are relatively small and the cost of DFT computations during training is acceptable, previous methods introducing self-consistency might outperform ours (as suggested by DEQHNet's results on water, where DEQHNet requires more data). However, when the dataset is large enough, or when it includes large molecules where DFT computation cost is prohibitive, the benefits of the DEQH model become more significant. As mentioned in Appendix A.11, since these methods are orthogonal, combining them could be a promising direction for future exploration.
>
> We will make sure to discuss these points more clearly in the revised manuscript.
>
> ## Questions:
> Thank you for your insightful question. Compared to works introducing self-consistency, our method enhances the model architecture. As mentioned in the related works section of our paper, several methods introduce this attribute during the training process at the loss level, requiring direct DFT computation. In contrast, the DEQH model does not directly involve DFT computation, leading to more efficient computational performance. Empirically, if the data volume and molecular size are relatively small, and the DFT computational cost during training is acceptable, previous works introducing self-consistency might perform better (as suggested by DEQHNet results on water, where DEQHNet requires a larger data volume). However, when the data volume is sufficient, or the data contains larger molecules making the DFT computation cost too high, the improvements offered by the DEQH model become more substantial. As we mentioned in Appendix A.11, since these methods are orthogonal, combining them is a direction worth exploring.

---

> > ### Comment · Reviewer_DtX8 · 2024-08-13
> >
> > I thank the author’s for the further explanation and clarifications. I keep my recommendation for acceptance with the score of a 7, conditioned on including discussions and additional experiments from the other rebuttals in the final manuscript.

---

> > > ### Author Response · Authors · 2024-08-13
> > >
> > > We are delighted that our response has met with your satisfaction. We will make sure to incorporate all of the latest discussions and experimental results into the manuscript and enhance its overall presentation promptly once the opportunity for a final polish is granted. Your constructive feedback is highly appreciated.

---

### Official Review · Reviewer_Tf4v · 2024-07-13

**Soundness:** 2
**Presentation:** 3
**Contribution:** 2
**Rating:** 4
**Confidence:** 2

**Summary:**

The authors introduce DEQH model, which combines deep equilibrium models with existing ML models to predict quantum Hamiltonian, and the author adopt QHNet as the backbone and further develop DEQHNet.  The authors evaluate the proposed method on benchmarks MD17 and QM9, the results show some effectiveness.

**Strengths:**

1. The idea of incorporating the DEQs is interesting, which have some kind of intrinsic principles as the authors demonstrated in the paper.

2. When compared with QHNet, the DEQHNet shows advantages.

**Weaknesses:**

1. Though I am relatively familiar with ML for Hamiltonian prediction and have gained some knowledge of DEQ, it is still not easy for me to understand the main part of this paper.

2. This work only compare with one baseline named QHNet, there are some other works which can also be baselines [1] [2] [3].

3. Source code is not available.

[1] Unke O, Bogojeski M, Gastegger M, et al. SE (3)-equivariant prediction of molecular wavefunctions and electronic densities[J]. Advances in Neural Information Processing Systems, 2021, 34: 14434-14447.

[2] Gong X, Li H, Zou N, et al. General framework for E (3)-equivariant neural network representation of density functional theory Hamiltonian[J]. Nature Communications, 2023, 14(1): 2848.

[3] Wang Y, Li H, Tang Z, et al. DeepH-2: Enhancing deep-learning electronic structure via an equivariant local-coordinate transformer[J]. arXiv preprint arXiv:2401.17015, 2024.

**Questions:**

1. Why is the model performance so much worse than the baseline in terms of energy? Has the author explored the reasons?

2. Is the method proposed by the author extensible and can it be applied to other existing ML models?

**Limitations:**

The authors have discussed the limitations.

---

> ### Author Rebuttal · Authors · 2024-08-05
>
> Thanks for your review. We will address each point in our response accordingly.
>
> ## Weakness 1:
> Thank you for your feedback. The DEQH model is designed to predict the Hamiltonian. Given the self-consistent iterative properties of the Hamiltonian, we regard these properties as a fixed-point iteration problem, which inspires us to use DEQ to introduce self-consistency into existing models for Hamiltonian prediction.
>
> We provide a PDF document in the global rebuttal section, which includes a figure delineating the distinction between the off-the-shelf model for Hamiltonian prediction and the DEQH model. Furthermore, we will revise our manuscript to simplify the technical language and include more background information about DEQs and Hamiltonian prediction. We will also add further explanations about the integration of these two components and how this combination contributes to the overall model. We hope that these revisions will make our paper more comprehensible to readers with various backgrounds.
>
> We appreciate your feedback and will make sure to address this concern in the revised version of our paper.
>
> ## Weakness 2:
> Thanks for your insightful suggestion. Although DEQH model could be extended to periodic systems, DEQHNet, being built on the foundation of QHNet, is tailored for non-periodic systems. Hence, a direct comparison with DeepH-E3 and DeepH-2 may not be suitable.
>
> In response to your feedback, we include additional benchmark results for SchNOrb [1], PhiSNet [2], and DeepH [3] on the MD17 dataset as follows.
>
> |Dataset|Model|H[10$^{-6} E_h$] $\downarrow$|$\epsilon$ [$10^{-6}E_h$] $\downarrow$|$\psi$ [$10^{-2}$] $\uparrow$|
> |--|--|--|--|--|
> |Water|SchNOrb|165.4|279.3|100.00|
> ||QHNet|10.79|33.76|99.99|
> ||PhiSNet (ori)|17.59|85.53|100.00|
> ||PhiSNet (reproduce)|15.67|-|99.94|
> ||DeepH|38.51|-|-|
> ||DEQHNet|36.07|335.86|99.99|
> |Ethanol|SchNOrb|187.4|334.4|100.00|
> ||QHNet|20.91|81.03|99.99|
> ||PhiSNet (ori)|12.15|62.75|100.00|
> ||PhiSNet (reproduce)|20.09|102.04|99.81|
> ||DeepH|22.09|-|-|
> ||DEQHNet|18.73|106.94|100.00|
> |Malonaldehyde|SchNOrb|191.1|400.6|99.00|
> ||QHNet|21.52|82.12|99.92|
> ||PhiSNet (ori)|12.32|73.50|100.00|
> ||PhiSNet (reproduce)|21.31|100.60|99.89|
> ||DeepH|20.10|-|-|
> ||DEQHNet|17.97|93.79|99.90|
> |Uracil|SchNOrb|227.8|1760|90.00|
> ||QHNet|20.12|113.44|99.89|
> ||PhiSNet (ori)|10.73|84.03|100.00|
> ||PhiSNet (reproduce)|18.65|143.36|99.86|
> ||DeepH|17.27|-|-|
> ||DEQHNet|15.07|107.49|99.89|
>
> However, it should be noted that the data used in DeepH is re-labeled by OpenMX[4]. Furthermore, considering the high training cost of the original PhiSNet, we provide PhiSNet results from the QHNet paper for a more balanced comparison.
>
> We appreciate your suggestion and will make sure to incorporate these benchmark results in the revised version of our paper.
>
> [1] Schütt K T, Gastegger M, Tkatchenko A, et al. Unifying machine learning and quantum chemistry with a deep neural network for molecular wavefunctions[J]. Nature communications, 2019, 10(1): 5024.
>
> [2] Unke O, Bogojeski M, Gastegger M, et al. SE (3)-equivariant prediction of molecular wavefunctions and electronic densities[J]. Advances in Neural Information Processing Systems, 2021, 34: 14434-14447.
>
> [3] Li H, Wang Z, Zou N, et al. Deep-learning density functional theory Hamiltonian for efficient ab initio electronic-structure calculation[J]. Nature Computational Science, 2022, 2(6): 367-377.
>
> [4] Ozaki T, Kino H. Numerical atomic basis orbitals from H to Kr[J]. Physical Review B, 2004, 69(19): 195113.
>
> ## Weakness 3:
> Thanks for your feedback. We understand the importance of making source code available for reproducibility and transparency in research. We have prepared an anonymous link (https://anonymous.4open.science/r/nips-rebuttal-80C4/) for the review.
>
> ## Question 1:
> Thanks for your question. In Section A.9 of Supplementary Material, we discussed this difference in detail. We randomly selected a molecule, added a symmetrized (Hamiltonian matrices need to be symmetric) Gaussian noise matrix, and solved the corresponding generalized eigenvalue equation. As shown in Fig. 5, when the noise on the Hamiltonian gradually increased, the error range of orbital energy also increased. The orbital energy errors we reported for both QHNet and DEQHNet on the MD17 and QH9 datasets fall within the range shown in the figure, which suggests that the experimental results are reasonable. This observation implies that only when the Mean Absolute Error (MAE) of the Hamiltonian is sufficiently small can we ensure that the corresponding orbital energy deviation is also small. Additionally, we repeated the experiments on the MD17 dataset and found that when the Hamiltonian MAE of the model is at the same level, it is possible to obtain both a lower and a higher orbital energy MAE than the baseline. In the comparison between QHNet and PhiSNet, we observed similar situations. This somewhat suggests that using existing Hamiltonian error measurements cannot definitively reflect the errors in the eigenvalues. This is a nontrivial, separate research question that we plan to investigate in the future.
>
> We will elaborate on this point more clearly in the revised version of our paper to provide a comprehensive understanding of the model's performance.
>
> ## Question 2:
> Thanks for your question. Indeed, our approach is designed to be highly adaptable and can be applied to off-the-shelf machine learning models. As stated in the introduction, our method introduces self-consistency to off-the-shelf models used for Hamiltonian prediction through the use of Deep Equilibrium (DEQ) models. To incorporate this into existing models, one simply needs to add a block that processes node features constructed from the Hamiltonian and overlap matrix inputs. This makes it relatively straightforward to combine any existing model with DEQ.
>
> We believe this adaptability is one of the key strengths of our method and will further highlight this in our revised manuscript.

---

> > ### Comment · Reviewer_Tf4v · 2024-08-10
> >
> > Thank you for your efforts and detailed response. However, I still have some concerns about the additional experiments: DEQHNet performed poorly on water, not even surpassing QHNet; its performance on ethanol was also not better than QHNet; and on Malonaldehyde, it showed no advantage compared to the baseline. It only had some advantage on uracil.
> >
> > Additionally, although the authors provided visualizations to demonstrate the method, I still believe that the presentation of the paper needs further improvement (the Reviewer eJGc also mentioned this point).
> >
> > In summary, although the idea of DEQHNet seems interesting, its presentation and experimental performance are not convincing enough, so I will maintain my original rating.

---

> ### Author Response · Authors · 2024-08-10
>
> Thank you for sharing your further comment. Nevertheless, we are afraid that your comment is based on a factual misunderstanding. The fact is, DEQHNet outperforms QHNet in all cases, including ethanol and malonaldehyde, with the only exception of water due to limited available labels (500; other molecules have 25,000 labels) as we have explained in the paper. Such results indicate that DEQHNet improves upon existing end-to-end Hamiltonian prediction methods universally.
>
> We are not sure about the specific reason why you have the misunderstanding, but we would like to stress that the primary comparing metric should be Hamiltonian MAE, which directly measures the prediction accuracy of the models. Other metrics, i.e., orbital energy and orbital coefficients, are derived quantities from the Hamiltonian. As we mentioned in the rebuttal, the prevailing MAE metric for Hamiltonian has been found not perfectly aligning with metrics of these derived quantities, which has also been observed in the comparison between QHNet and PhiSNet. In the appendix of our paper and in the rebuttal, we have explained why our method performs well in terms of Hamiltonian error and orbital coefficients. We also pointed out that the reported orbital energy falls within the error range of the current Hamiltonian. Specifically, in Section A.9 of the Supplementary Material and the Question 1 section of the rebuttal, we discussed this discrepancy in detail. We randomly selected a molecule, added a symmetrized Gaussian noise matrix (Hamiltonian matrices need to be symmetric), and solved the corresponding generalized eigenvalue equation. As shown in Fig. 5, as the noise on the Hamiltonian increased, the error range of the orbital energy also increased. The orbital energy errors we reported for both QHNet and DEQHNet on the MD17 and QH9 datasets fall within the range shown in the figure, suggesting that the experimental results are reasonable.
>
> Additionally, we can provide further evidence from the QHNet paper, which observed a similar phenomenon. In their study, the PhiSNet achieved a Hamiltonian error of only 15.67x10$^{-6}$  Hartree on water, but the orbital energy error was "ten times worse than that by QHNet (33.76)", approximately  330x10$^{-6}$  Hartree.
>
> This observation implies that only when the Mean Absolute Error (MAE) of the Hamiltonian is sufficiently small can we ensure that the corresponding orbital energy deviation is also small. In the comparison between QHNet and PhiSNet, we observed similar situations. This somewhat suggests that using existing Hamiltonian error measurements cannot definitively reflect the errors in the eigenvalues. These evidence suggests that it is a nontrivial and long-standing challenge  in the domain of Hamiltonian prediction to design a proper metric on the matrix space so that reflects the metric of derived quantities e.g. energy, for which we will investigate in future work.
>
> Finally, concerning your comment that "the presentation of the paper needs further improvement," we have already stated in our rebuttal that "We will also add further explanations about the integration of these two components and how this combination contributes to the overall model. We hope that these revisions will make our paper more comprehensible to readers with various backgrounds." The PDF we provided in the global rebuttal is merely an initial visualization, as the current NeurIPS review process only permits a one-page PDF that includes images and tables but no text.

---

### Official Review · Reviewer_eJGc · 2024-07-30

**Soundness:** 3
**Presentation:** 3
**Contribution:** 3
**Rating:** 6
**Confidence:** 2

**Summary:**

This paper introduces the DEQH (Deep Equilibrium Quantum Hamiltonian) model, which integrates Deep Equilibrium Models (DEQs) for predicting quantum Hamiltonians. By incorporating DEQs, the model captures the self-consistency of Hamiltonians without needing iterative Density Functional Theory (DFT) calculations during training, enhancing computational efficiency. DEQHNet, a specific implementation, demonstrates improvements in prediction accuracy on the MD17 and QM9 datasets. The model acts as both a predictor and solver, iteratively refining the Hamiltonian to achieve self-consistency. Ablation studies further validate the effectiveness of this approach.

**Strengths:**

1. The DEQH model proposed eliminates the need for iterative DFT calculations during training, which reduces computational overhead.

2. The DEQHNet model demonstrates improved accuracy in predicting Hamiltonians on the MD17 and QM9 datasets.

3. The model inherently captures the self-consistency required for accurate Hamiltonian prediction.

4. This paper includes ablation studies to analyze the contribution of different components of the model.

5. The paper shows quick convergence of the DEQHNet model.

**Weaknesses:**

1. The presentation of the paper could be improved. In this paper, the integration of Deep Equilibrium Models (DEQs) with the Hamiltonian solver is presented in a quite technical way. Additional visual illustrations and diagrams could help clarify the workflow of the DEQH model and its components.

2. This paper primarily compares DEQHNet with QHNet. Benchmarking with additional methods could help better evaluate the results.

**Questions:**

What are the runtime and memory usage for DEQHNet compared to previous methods?

What are the theoretical assumptions underlying the DEQH model, and how do they impact the generalizability of the results?

**Limitations:**

The limitations are discussed.

---

> ### Author Rebuttal · Authors · 2024-08-05
>
> We are immensely grateful for your assessments. We will address each point in our response accordingly.
>
> ## Weakness 1:
> Thanks for your feedback and suggestions. We provide a PDF document in the global rebuttal section, which includes a figure delineating the distinction between the off-the-shelf model used for Hamiltonian prediction and the DEQH model. We hope this visual aid will enable readers to gain a clearer and more intuitive understanding.
>
> Furthermore, in the revised version, we will include additional diagrams and illustrations to better depict the workflow of the DEQH model and its components. This will include detailed flowcharts and schematic representations that highlight the key processes and interactions within the model. We are grateful for your insights, which have been instrumental in improving the clarity and quality of our work.
>
> ## Weakness 2:
> Thank you for your valuable feedback. We agree that adding more benchmark results would provide a more comprehensive evaluation of our DEQHNet model's performance. In response to your suggestion, we include additional benchmark results for SchNOrb [1], PhiSNet [2], and DeepH [3] on the MD17 dataset as follows.
>
> |Dataset|Model|H[10$^{-6} E_h$] $\downarrow$|$\epsilon$ [$10^{-6}E_h$] $\downarrow$|$\psi$ [$10^{-2}$] $\uparrow$|
> |--|--|--|--|--|
> |Water|SchNOrb|165.4|279.3|100.00|
> ||QHNet|10.79|33.76|99.99|
> ||PhiSNet (ori)|17.59|85.53|100.00|
> ||PhiSNet (reproduce)|15.67|-|99.94|
> ||DeepH|38.51|-|-|
> ||DEQHNet|36.07|335.86|99.99|
> |Ethanol|SchNOrb|187.4|334.4|100.00|
> ||QHNet|20.91|81.03|99.99|
> ||PhiSNet (ori)|12.15|62.75|100.00|
> ||PhiSNet (reproduce)|20.09|102.04|99.81|
> ||DeepH|22.09|-|-|
> ||DEQHNet|18.73|106.94|100.00|
> |Malonaldehyde|SchNOrb|191.1|400.6|99.00|
> ||QHNet|21.52|82.12|99.92|
> ||PhiSNet (ori)|12.32|73.50|100.00|
> ||PhiSNet (reproduce)|21.31|100.60|99.89|
> ||DeepH|20.10|-|-|
> ||DEQHNet|17.97|93.79|99.90|
> |Uracil|SchNOrb|227.8|1760|90.00|
> ||QHNet|20.12|113.44|99.89|
> ||PhiSNet (ori)|10.73|84.03|100.00|
> ||PhiSNet (reproduce)|18.65|143.36|99.86|
> ||DeepH|17.27|-|-|
> ||DEQHNet|15.07|107.49|99.89|
>
> We would like to note that due to the design of DeepH, the data used is re-labeled by OpenMX [4], which may lead to different results compared to other models. Furthermore, considering the high training cost of PhiSNet, we also provide PhiSNet results from the QHNet article for a more balanced and comprehensive comparison.
>
> These results will be incorporated into the revised manuscript to provide a more robust and thorough evaluation of DEQHNet, thereby offering readers a clearer understanding of its performance relative to other state-of-the-art methods.
>
> [1] Schütt K T, Gastegger M, Tkatchenko A, et al. Unifying machine learning and quantum chemistry with a deep neural network for molecular wavefunctions[J]. Nature communications, 2019, 10(1): 5024.
>
> [2] Unke O, Bogojeski M, Gastegger M, et al. SE (3)-equivariant prediction of molecular wavefunctions and electronic densities[J]. Advances in Neural Information Processing Systems, 2021, 34: 14434-14447.
>
> [3] Li H, Wang Z, Zou N, et al. Deep-learning density functional theory Hamiltonian for efficient ab initio electronic-structure calculation[J]. Nature Computational Science, 2022, 2(6): 367-377.
>
> [4] Ozaki T, Kino H. Numerical atomic basis orbitals from H to Kr[J]. Physical Review B, 2004, 69(19): 195113.
>
> ## Question 1:
> Thanks for your question regarding the runtime and memory usage of DEQHNet compared to previous methods. As discussed in Section A.10 of our Supplementary Material, the training time of DEQHNet on the QH9 dataset is approximately 1.5 times that of QHNet. This is primarily due to the need for DEQ models to perform fixed-point iteration solving, which results in a higher runtime compared to models that do not incorporate a self-consistency mechanism.
>
> Regarding memory usage, DEQHNet requires extra blocks to process node features constructed from the Hamiltonian and overlap matrix inputs. This results in a slightly larger memory footprint compared to previous network structures. In our experiments, DEQHNet only had a 28.5% increase in parameters compared to QHNet, demonstrating that it improved accuracy without a significant increase in parameters.
>
> We are currently quantifying these requirements more precisely and will include a detailed comparison of memory usage in the revised manuscript.
>
> ## Question 2:
> Thanks for your insightful question. The theoretical foundations of our DEQH model are rooted in the common practice in Density Functional Theory (DFT) of iteratively solving the electronic Schrödinger equation. The process begins by initializing a Hamiltonian, solving the generalized eigenvalue problem, and then constructing a new Hamiltonian for the next iteration. We treat this iterative property of the Hamiltonian as a fixed-point iteration problem. This allows us to leverage the characteristics of DEQ in solving fixed-point problems to impose self-consistency on the network for Hamiltonian prediction.
>
> The results from the QH9-stable-ood experiment demonstrate that our approach can significantly reduce the Mean Absolute Error (MAE) of the Hamiltonian, suggesting an enhanced capability for generalization. Additionally, the DFT Acceleration ratio presented in Appendix A.8 further reflects the generalizability of our model to a certain extent. We hypothesize that introducing DEQ enables the network to learn the iterative mechanism of solving the Hamiltonian, rather than directly learning the solution to the Hamiltonian. This shift could allow the network to better capture inherent physical laws, thereby improving generalizability.

---

> > ### Comment · Reviewer_eJGc · 2024-08-11
> >
> > Thanks a lot for taking the time and effort to answer my questions. I would be considering raising the score in the next two days.

---

> > > ### Author Response · Authors · 2024-08-12
> > >
> > > We are glad to know that our response is satisfactory to you. We will make sure to include all the new experimental results in the manuscript and improve the presentation as soon as we are permitted a polish for the final version. We are grateful for your valuable feedback.

---

> > > > ### Comment · Reviewer_eJGc · 2024-08-13
> > > >
> > > > I just raised my score. Thanks!

---

> > > > > ### Author Response · Authors · 2024-08-14
> > > > >
> > > > > We are very grateful for the increased score and deeply appreciate your valuable feedback.

---

### Author Rebuttal · Authors · 2024-08-05

We are grateful for the valuable feedback provided by all the reviewers. We provide a PDF document in the global rebuttal section, which includes a figure delineating the distinction between the off-the-shelf model used for Hamiltonian prediction and the DEQH model. We hope this visual aid will enable readers to gain a clearer and more intuitive understanding.

---

### Decision · Program_Chairs · 2024-09-25

**Decision:**

Accept (poster)

**Comment:**

The paper introduces a novel neural architecture for quantum Hamiltonian prediction based on deep equilibrium models to model iterative algorithms. Benchmarks against competing methods show in general improvements in the MAE on Hamiltonian predictions.
Some concerns highlighted by the reviews are around clarity of the presentation and the fact that better Hamiltonian prediction does not translate to better prediction of other molecular properties. The authors addressed these providing further experimental results during the discussion phase that helped understanding the performance of the model.